# ONE SCALE AT A TIME: SCALE-AUTOREGRESSIVE MODELING FOR FLUID FLOW DISTRIBUTIONS

## ABSTRACT

Analyzing unsteady fluid flows often requires access to the full distribution of possible temporal states, yet conventional PDE solvers are computationally prohibitive and learned time-stepping surrogates quickly accumulate error over long rollouts. Generative models avoid compounding error by sampling states independently, but diffusion and flow-matching methods, while accurate, are limited by the cost of many evaluations over the entire mesh. We introduce *scale–autoregressive modeling* (SAR) for sampling flows on unstructured meshes hierarchically from coarse to fine: it first generates a low-resolution field, then refines it by progressively sampling higher resolutions conditioned on coarser predictions. This coarse-to-fine factorization improves efficiency by concentrating computation at coarser scales, where uncertainty is greatest, while requiring fewer steps at finer scales. Across unsteady-flow benchmarks of varying complexity, SAR attains substantially lower distributional error and higher per-sample accuracy than state-of-the-art diffusion models based on multi-scale GNNs, while matching or surpassing a flow-matching Transolver (a linear-time transformer) yet running 2–7× faster than this depending on the task. Overall, SAR provides a practical tool for fast and accurate estimation of statistical flow quantities (e.g., turbulent kinetic energy and two-point correlations) in real-world settings.[1]

## 1 INTRODUCTION

Fluid dynamics plays a central role in a wide range of scientific and engineering fields, including aerospace design (Moxey et al., 2020; Jané-Ippel et al., 2023), civil infrastructure (Zheng & Zhang, 2012), biomedical applications (Doost et al., 2016; Peiffer et al., 2013), and computer graphics (Bridson, 2015). Traditionally, fluid behavior is modeled by numerically solving partial differential equations (PDEs). While models like Reynolds-averaged Navier-Stokes (RANS) offer coarse estimates of mean flows (Alfonsi, 2009), many real-world flows exhibit complex unsteady dynamics that require access to full state distributions over time to be properly described, for instance, via statistical measures such as root-mean-square (RMS) fluctuations and two-point correlations (Pope, 2000; Wilcox, 1998). Capturing these distributions typically demands long and computationally expensive simulations, especially in 3D turbulent regimes (Caros et al., 2022).

Advances in deep learning have enabled surrogate models that learn the temporal evolution of physical systems from data (Kim et al., 2019; Stachenfeld et al., 2021; Pfaff et al., 2021). However, these models often degrade over long time horizons due to error accumulation during iterative rollout (Kohl et al., 2024). In contrast, generative modeling provides an alternative for capturing fully developed flow distributions without relying on time-marching (Lienen et al., 2024; Lino et al., 2025). These models learn the underlying data distribution and can generate converged flow states directly, conditioned on domain geometry and boundary conditions—bypassing the need to simulate the transient warm-up phase. By drawing multiple samples, one can estimate statistical flow quantities, and because each sample is generated independently, error does not accumulate over time. Among generative methods, diffusion (including flow-matching) models have demonstrated superior sample fidelity and distributional accuracy (Dhariwal & Nichol, 2021b; Liu & Thuerey, 2024; Lino et al., 2025). However, their practical deployment is limited by high computational cost: each sample requires dozens of denoising steps, and accurate transformer-based models further increase the burden due to the global receptive field brought by attention mechanisms (Peebles & Xie, 2023).

---

[1]Code is available at on-acceptance.

To address these challenges, we introduce *scale-autoregressive modeling* (SAR), a generative framework designed for fluid domains with general geometries and unstructured discretizations. SAR generates physical fields hierarchically, proceeding autoregressively from coarse to fine spatial resolutions (Figure 1a). At each step, SAR first computes a contextual representation of the previously generated coarser scales, which then conditions a small diffusion model to generate the solution at the next finer scale. This hierarchical formulation enables to assign denoising steps adaptively: coarser scales, which carry higher uncertainty, receive more steps, while finer scales require fewer due to stronger conditioning. Since only a small number of steps are needed at high-resolution scales, SAR can incorporate attention layers for global context without incurring the cost of conventional diffusion transformer models.

We evaluate SAR on several unsteady fluid dynamics benchmarks, including pressure prediction on 3D wings in turbulent flow. Our results show that SAR outperforms state-of-the-art diffusion models based on multi-scale graph neural networks (GNNs) (Lino et al., 2025), and match the superior performance of a transformer-based diffusion model at a fraction of the computational cost.

## 2 RELATED WORK

**Probabilistic Modeling of Fluid Flows** Probabilistic models such as variational autoencoders (VAEs) (Kingma & Welling, 2014) and generative adversarial networks (GANs) (Goodfellow et al., 2014) have enabled modeling probability distributions over plausible physical states (Maulik et al., 2020; Drygala et al., 2022; Kim & Lee, 2020), but often struggle with complex multimodal distributions (Lino et al., 2025). Recently, denoising diffusion probabilistic models (DDPMs) and flow-matching models have emerged as powerful alternatives (Ho et al., 2020; Nichol & Dhariwal, 2021; Dhariwal & Nichol, 2021a; Lipman et al., 2023), with successful applications in flow-field super-resolution (Shu et al., 2023; Li et al., 2023b), uncertainty quantification (Liu & Thuerey, 2024), and improving stability of long-term simulations (Lippe et al., 2024; Rühling Cachay et al., 2024; Kohl et al., 2024). Closer to our work, Lienen et al. (2024) and Baldan et al. (2025) modeled the distribution of unsteady fully-developed flow solutions on structured grids, and Lino et al. (2025) extended this to unstructured meshes using multi-scale GNNs in latent spaces. While previous work applies all denoising steps to representations of fixed resolution, SAR departs from this approach by autoregressively generating the solution scale-by-scale (from coarser to finer levels), thereby avoiding the computational burden of full-resolution evaluations at every step and allowing the use of fewer denoising steps at finer scales, as illustrated in Figure 7.

**Learning Fluid Dynamics on General Geometries** To handle fluid domains with irregular geometries and enable adaptive spatial resolutions, GNNs (Pfaff et al., 2021; Lino et al., 2022) and transformers (Alkin et al., 2024; Wu et al., 2024) have emerged as prominent architectures. GNNs encode mesh information in graphs, while transformers process mesh nodes using spatial coordinates as inputs, though their time complexity scales non-linearly with node count. Recent variants mitigate this inefficiency by operating in fixed-size latent spaces (Alkin et al., 2024; 2025; Wen et al., 2025), using compact learned representations for each attention head (Wu et al., 2024; Luo et al., 2025), or adopting—often less accurate—linear attention (Hao et al., 2023; Li et al., 2022). Fluid flows involve highly non-local physics. While, multi-scale GNNs leverage hierarchical structures to capture non-local interactions (Lino et al., 2022; Fortunato et al., 2022; Cao et al., 2023), each individual attention layer in transformers inherently has a global receptive field. This property is particularly beneficial for diffusion models (Peebles & Xie, 2023), but it introduces significant computational overhead, even when using linear attention variants (Katharopoulos et al., 2020; Cao, 2021). Our SAR model strategically employs the *Transolver* transformer (Wu et al., 2024; Luo et al., 2025) within a hierarchical framework, selectively processing subsets of nodes to maintain computational efficiency while harnessing global spatial context.

**Autoregressive Image Generation** Autoregressive modeling, popularized by large language models (LLMs) (Vaswani et al., 2017; Radford et al., 2019), has also been adapted to image generation, with early work predicting tokens sequentially in raster-scan order (Razavi et al., 2019; Esser et al., 2021; Lee et al., 2022). Recent *masked-prediction* models have improved scalability by predicting multiple tokens per autoregressive step (He et al., 2022; Chang et al., 2022; Li et al., 2023a). Despite these advances, autoregressive models typically underperform diffusion models due to inadequate inductive biases. Notably, the arbitrary raster-scan order of token generation poorly reflects the spatial structure of images. Tian et al. (2024) addressed this by introducing coarse-to-fine au-

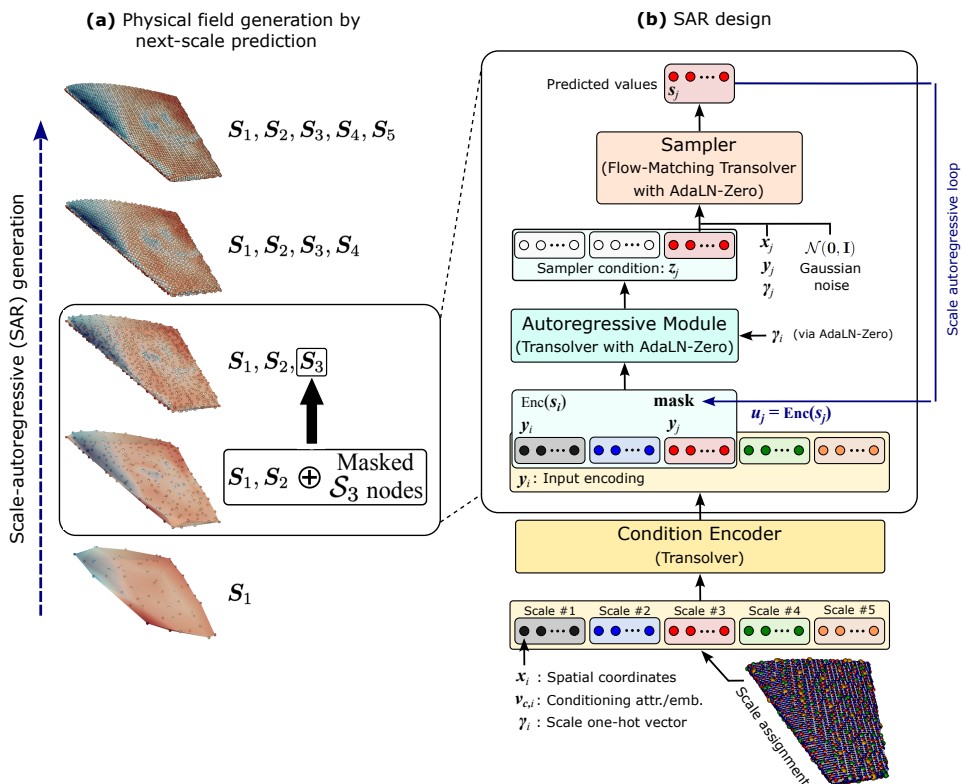

Figure 1: (a) SAR generates resolution scales autoregressively from coarser to finer. (b) A SAR model is consists of a condition encoder, an autoregressive module, and a flow-matching sampler.

toregressive modelling, using a multi-scale tokenizer that encodes images into hierarchical tokens at multiple resolutions, significantly enhancing image quality. Another limitation inherited from LLMs is the reliance on discrete embeddings. Li et al. (2024) showed that continuous embeddings can also be modeled autoregressively by using a small diffusion model to generate continuous values conditioned on deterministic outputs from the autoregressive transformer backbone. Inspired by these developments, our SAR approach introduces a hierarchical autoregressive method tailored for physical modeling on unstructured meshes. Moreover, while previous models sample multiple token embeddings independently at each autoregressive step (Tian et al., 2024; Li et al., 2024), SAR employs a transformer-based diffusion sampler that accounts for global spatial dependencies, leading to significantly improved sample quality.

## 3 METHOD

We introduce *scale–autoregressive modeling* (SAR) for efficiently sampling physical systems from their spatial discretization and governing parameters. SAR generates fields across a coarse-to-fine hierarchy (Figure 1a), focusing computation on coarser scales for faster and accurate sampling.

### 3.1 LEARNING DISTRIBUTIONS OF STATISTICALLY STATIONARY UNSTEADY FLOWS

We consider fluid domains of arbitrary geometry, each discretized using a mesh defined by a set of nodes $\mathcal{V}_M$ and edges $\mathcal{E}_M$. Every node $i \in \mathcal{V}_M$ is associated with a spatial position $\boldsymbol{x}_i \in \mathbb{R}^d$. The system's state at time $t$ is described by $F$ continuous fields (e.g., velocity components and pressure), sampled at the mesh nodes. Fluid systems typically exhibit transient behavior before reaching a statistically stationary regime. In this regime, individual realizations may still display chaotic or oscillatory dynamics, but statistical quantities—such as variances and spatial correlations— converge to stable values (Wilcox, 1998; Pope, 2000). Crucially, these stationary dynamics depend only on the domain geometry and governing physical parameters, and not on initial conditions.

Our objective is to learn a generative model capable of directly sampling from the equilibrium distribution, thus bypassing costly simulations of the transient phase. By generating multiple samples, we can approximate the stationary flow distribution and compute relevant statistical measures.

## 3.2 SCALE AUTOREGRESSIVE MODELING

### 3.2.1 NEXT SCALE PREDICTION

The system's conditioning information is represented in a directed graph $\mathcal{G} := (\mathcal{V}, \mathcal{E})$, where $\mathcal{V} \equiv \mathcal{V}_M$ corresponds to the set of mesh nodes and $\mathcal{E}$ denotes a set of bi-directional edges derived from the mesh edges $\mathcal{E}_M$. Node attributes $\boldsymbol{V}_c := \{\boldsymbol{v}_i^c \mid i \in \mathcal{V}\}$ encode problem-specific conditioning features, such as the Reynolds number ($Re$). Edge attributes $\boldsymbol{E}_c := \{\boldsymbol{e}_{ij}^c \mid (i,j) \in \mathcal{E}\}$ represent relative positions between nodes (i.e., $\boldsymbol{x}_j - \boldsymbol{x}_i$). While these are not used by the main SAR components, they are utilized by the VAE in the latent variant of our model, which we adopt.

To model spatial hierarchy, SAR partitions the node set $\mathcal{V}$ into $K$ disjoint subsets $\mathcal{S}_1, \mathcal{S}_2, \ldots, \mathcal{S}_K$, each corresponding to a resolution scale, with $|\mathcal{S}_1| < |\mathcal{S}_2| < \cdots < |\mathcal{S}_K|$. Coarser scales yield compact representations of the domain, while finer scales include more nodes and capture greater physical detail. This hierarchy is constructed using a multigrid coarsening algorithm (Guillard, 1993), and assigning a unique scale to each node, as outlined in Algorithm 1. Given these scales, SAR formulates generative modeling as a *next-scale prediction* task. At autoregressive step $k$, it generates field values for all the nodes in $\mathcal{S}_k$, conditioned on all coarser-scale predictions and the system's geometric and physical characteristics. The joint likelihood is factorized as

$$p(\boldsymbol{S}_{1:K}) = \prod_{k=1}^{K} p(\boldsymbol{S}_k \mid \boldsymbol{X}, \boldsymbol{V}_c, \Gamma, \boldsymbol{S}_{1:k-1}), \tag{1}$$

where $\boldsymbol{S}_{1:k} := \{\boldsymbol{S}_1, \boldsymbol{S}_2, \ldots, \boldsymbol{S}_k\}$, $\boldsymbol{S}_k := \{\boldsymbol{s}_i \in \mathbb{R}^F \mid i \in \mathcal{S}_k\}$ denotes the value of the physical fields at scale-$k$ nodes, $\boldsymbol{X} = \{\boldsymbol{x}_i \in \mathbb{R}^d \mid i \in \mathcal{V}\}$ denotes the spatial coordinates of all nodes, and $\Gamma = \{\gamma_i \in \mathbb{N} \mid i \in \mathcal{V}\}$ indicates the nodes' scale.

The $k$-th autoregressive step in SAR samples $\boldsymbol{S}_k$ from a learned approximation of $p(\boldsymbol{S}_k \mid \boldsymbol{X}, \boldsymbol{V}_c, \Gamma, \boldsymbol{S}_{1:k-1})$. This is achieved through two sequential subprocesses: first, computing how the coarser scales condition the next one; and second, sampling the solution at the next scale conditioned on this information. The first subprocess integrates information from all coarser-scale predictions $\boldsymbol{S}_{1:k-1}$ to construct a latent representation, $\boldsymbol{Z}_k := \{\boldsymbol{z}_j \mid j \in \mathcal{S}_k\}$, for each node in the next finer scale. This representation is then passed to a diffusion-based sampler, which generates the physical fields at the next scale conditioned on it. This strategy enables the use of a different number of denoising steps per scale, depending on the level of uncertainty, without requiring scale-specific modules—since all model components are shared across scales. Specifically, to balance computational cost and accuracy, SAR allocates more denoising steps to coarser scales, where uncertainty is typically higher, and fewer to finer scales, which benefit from stronger conditioning.

### 3.2.2 SPECIALIZED SAR COMPONENTS

SAR is realized through three interdependent and specialized components: a condition encoder, an autoregressive module, and a sampler (Figure 1b). These components are described below.

**Condition Encoder** The condition encoder processes the entire node set $\mathcal{V}$ to embed its geometric and physical information into node-wise feature vectors. The input attributes for each node $i \in \mathcal{V}$ include its spatial coordinates $\boldsymbol{x}_i$, conditioning attributes $\boldsymbol{v}_i^c$, and a one-hot vector for the scale index $\gamma_i$. Its task is to aggregate these inputs into a latent representation $\boldsymbol{y}_i$ for each node, each of which individually captures both local features and global context across the domain. Formally, we define: $\boldsymbol{Y} = \text{CONDITIONENCODER}(\boldsymbol{X}, \boldsymbol{V}_c, \Gamma)$, where $\boldsymbol{Y} := \{\boldsymbol{y}_i \mid i \in \mathcal{V}\}$. These global encoding vectors ensure that during the subsequent autoregressive generation—where finer-scale information is not yet available—each node still retains access to the full geometric and physical context. Although this encoder must operate over the full set of nodes $\mathcal{V}$, it is evaluated only once per generated sample and can remain lightweight, as it is not responsible for probabilistic modeling. Moreover, when generating multiple samples for the same domain geometry and physical parameters—which is often the case when estimating statistics—$\boldsymbol{Y}$ can be cached, and the condition encoder needs to be evaluated only once. To efficiently process global interactions, we implement the condition encoder using the Transolver architecture proposed in Wu et al. (2024).

**Autoregressive Module** The autoregressive module is evaluated at the beginning of each autoregressive step. At step $k$, it processes the nodes in the target scale $\mathcal{S}_k$ along with all nodes from the coarser scales $\mathcal{S}_1, \ldots, \mathcal{S}_{k-1}$. Its objective is to determine how the global condition encodings $\boldsymbol{Y}_1, \boldsymbol{Y}_2, \ldots, \boldsymbol{Y}_k$ (where $\boldsymbol{Y}_l := \{\boldsymbol{y}_i \mid i \in \mathcal{S}_l\}$) and the autoregressively generated coarser-scale predictions $\boldsymbol{S}_1, \boldsymbol{S}_2, \ldots, \boldsymbol{S}_{k-1}$ influence the solution at the next scale $\mathcal{S}_k$. The output is a new latent representation, $\boldsymbol{Z}_k = \{\boldsymbol{z}_j \mid j \in \mathcal{S}_k\}$, for each node $j \in \mathcal{S}_k$. This is given by

$$\boldsymbol{Z}_k = \mathrm{AR}(k, \boldsymbol{Y}_{1:k}, \boldsymbol{S}_{1:k-1}). \tag{2}$$

We implement this module using a Transolver backbone, augmented with AdaLN-Zero blocks (Peebles & Xie, 2023) to condition both the attention and MLP layers on the current autoregressive step $k$. This is encoded as a learnable embedding, with a distinct vector assigned to each possible scale. The input feature for each node $i$ in a coarser scale $l < k$ is constructed by concatenating its condition encoding $\boldsymbol{y}_i$ with a linear projection of its already predicted field values $\boldsymbol{s}_i$. For nodes in the target scale $\mathcal{S}_k$, the input consists of their condition encoding $\boldsymbol{y}_i$ concatenated with a learnable *mask* embedding vector, indicating that their field values are yet to be predicted.

**Sampler** The sampler is the probabilistic model responsible for generating the output field values at each scale, conditioned on the nodes' spatial coordinates, scale $\gamma_j$, and the latent vectors $\boldsymbol{y}_j$ and $\boldsymbol{z}_j$. Formally, at autoregressive step $k$, the sampling process is defined as

$$\boldsymbol{S}_k \sim \mathrm{SAMPLER}(\boldsymbol{S}_k \mid \boldsymbol{X}_k, \boldsymbol{Y}_k, \boldsymbol{Z}_k, \Gamma_k). \tag{3}$$

Within the SAR model, the sampler is evaluated at the end of each autoregressive step, directly following the autoregressive module. While the sampler can, in principle, adopt any probabilistic modeling framework, we employ a diffusion-based approach due to its demonstrated effectiveness. In particular, we adopt a flow-matching formulation, which reduces the number of required denoising steps during inference (Lipman et al., 2023). The architecture of the sampler is also based on a Transolver backbone, augmented with AdaLN-Zero blocks, which condition the network on a sinusoidal embedding of the denoising-time (Vaswani et al., 2017; Peebles & Xie, 2023).

The number of denoising steps required by the sampler at each scale depends on the complexity of the conditional distribution in equation 3, influenced by factors such as multimodality and variability. The condition encoder and the autoregressive module provide the sampler—via $\boldsymbol{Y}_k$ and $\boldsymbol{Z}_k$, respectively—with both global context and coarser-scale predictions. As generation proceeds from coarser to finer scales, the conditioning becomes increasingly informative, as more of the hierarchy has already been predicted. This reduces output uncertainty at later steps. As a result, stochastic complexity is concentrated at earlier steps (i.e., coarser scales), which require more denoising steps, while finer scales can be processed with significantly fewer. Furthermore, since finer scales contain substantially more nodes, reducing the number of denoising steps at them significantly improves inference speed without sacrificing output quality.

**Latent-Space SAR** An additional strategy for reducing the required number of denoising steps at the final scale without compromising output quality is to apply SAR in the latent space of a separately trained VAE, rather than directly in physical space. This VAE is relatively compact, comprising only two message-passing layers in both the encoder and decoder, without node compression. To enhance robustness against latent-space noise, Gaussian noise (with a standard deviation of $10^{-2}$) is introduced during VAE training. At inference time, after the SAR model has finished generating the finest scale, the predicted node features $\boldsymbol{S}_{1:k}$ and the nodes' relative positions $\boldsymbol{E}_c$ are passed through the VAE decoder. This step effectively removes residual noise and, to some extent, corrects minor misalignments between scales.

### 3.2.3 SAR TRAINING OBJECTIVE

The full SAR model is trained by optimizing the flow-matching objective applied to the output of the sampler network. Let $q_{k,r}$ denote the probability path followed by the solution $\boldsymbol{S}_{k,r}$ through *denoising-time* $r$, where $q_{k,0}$ corresponds to a standard normal distribution, and $q_{k,1}$ approximates the distribution of the training data for $\boldsymbol{S}_k$. The flow-matching objective aims to match this target probability path (Lipman et al., 2023). Specifically, for a given scale $k$, we optimize

$$\mathcal{L}_k(\boldsymbol{X}, \boldsymbol{V}_c, \Gamma, \boldsymbol{S}_{1:k-1}) := \mathbb{E}_{r, \sim q_{k,r}(\boldsymbol{S}_{k,r})} \|\boldsymbol{w}_{k,r}(\boldsymbol{S}_{k,r}) - \boldsymbol{u}_{k,r}(\boldsymbol{S}_{k,r} \mid \boldsymbol{X}_k, \boldsymbol{Y}_k(\boldsymbol{X}, \boldsymbol{V}_c, \Gamma), \boldsymbol{Z}_k(\boldsymbol{Y}_{1:k}, \boldsymbol{S}_{1:k-1}), \Gamma_k)\|^2, \tag{4}$$

where $\boldsymbol{w}_{k,r}$ is the target flow vector field and $\boldsymbol{u}_{k,r}$ is its neural network approximation. Note that $\boldsymbol{Y}_k$ is modeled by the conditioning encoder, $\boldsymbol{Z}_k$ by the autoregressive module, and $\boldsymbol{u}_{k,r}$ by the sampler. The overall objective is to minimize the sum of losses over all scales, $\sum_k^K \mathcal{L}_k$.

During training, we randomly select a scale $k$ and a sample from the dataset (providing inputs $\boldsymbol{X}, \boldsymbol{V}_c, \Gamma$, and $\boldsymbol{S}_{1:k}$; and targets $\boldsymbol{S}_k$), and compute the corresponding loss $\mathcal{L}_k$. To stabilize convergence, for each training sample, we randomly draw four different values of $r \in [0, 1]$ from a uniform distribution and use them to evaluate the loss at multiple points along the denoising trajectory. Since autoregressive models are sensitive to error accumulation, we introduce Gaussian noise to the lower-resolution inputs $\boldsymbol{S}_{1:k-1}$ during training. This enhances robustness against small prediction errors at inference time (Sanchez-Gonzalez et al., 2020).

## 4 EXPERIMENTS

**Benchmarks**  We evaluate SAR on the three benchmark domains introduced in Lino et al. (2025) for probabilistic modeling of unsteady fluid dynamics on meshes: (i) wall pressure on an elliptical body in 2D quasi-periodic laminar flow (ELLIPSE); (ii) full-field velocity and pressure around the same geometry (ELLIPSEFLOW); and (iii) surface pressure on a wing in 3D turbulent flow (WING). ELLIPSEFLOW represents a canonical fluid scenario, with varying Reynolds numbers and aspect ratios. The WING datasets comprise turbulent flow simulations over wings with varying geometric parameters, including sweep, twist, taper ratio, and thickness. Modeling this flow regime is particularly challenging due to its chaotic, high-dimensional nature and multi-scale interactions. ELLIPSEFLOW includes refined meshes near the object boundary to better resolve near-wall features, while ELLIPSE and WING restrict supervision to the surface of the immersed object—showcasing the computational advantages of unstructured, surface-based representations.

**Baselines**  We compare SAR against state-of-the-art diffusion graph networks (DGNs), flow-matching graph networks (FM-GNNs), and their latent space variants (LDGN and LFM-GNN, respectively) (Lino et al., 2025). These models use a multi-scale GNN backbone to enable efficient denoising on large domains. We also compare SAR against a flow-matching Transolver (FMT) baseline that applies a Transolver network directly to the full set of nodes $\mathcal{V}$ for denoising. This setup can be considered equivalent to a single-scale, sampler-only SAR variant, and it is trained using the same strategy described in Section 3.2.3. Following Lino et al. (2025), all models are trained on only 10 consecutive states per system for the ELLIPSE and ELLIPSEFLOW domains (**26–48%** of the time points required to capture a full vortex-shedding cycle), and 250 consecutive states for the WING domain (**10%** of the time points needed to reach statistically stationary variance). This setup evaluates the ability to learn the full underlying distributions from short trajectories across different systems (i.e., different geometries and/or physical parameters). At inference time, all models apply equispaced denoising steps. Diffusion models follow the fast sampling strategy proposed by Song & Ermon (2020), as also adopted in Lino et al. (2025), while flow-matching models (including the SAR sampler) use forward Euler integration. Unless otherwise specified, SAR models are implemented with three scales, and the reported results use the number of denoising steps yielding the best or converged accuracy. Further experimental details are provided in Appendix A.3.

Table 1: Wasserstein-2 distance ($W_2$) on the ELLIPSEFLOW datasets.

| ELLIPSE FLOW / Model | -INDIST | -LOWRE | -HIGHRE | -THIN | -THICK | -AOA | #steps |
|---|---|---|---|---|---|---|---|
| DGN (Lino et al. 2025) | 4.72 ± 2.10 | 4.04 ± 1.74 | 5.48 ± 2.01 | 3.20 ± 0.81 | 7.76 ± 2.39 | 5.96 ± 2.12 | 50 |
| LDGN (Lino et al. 2025) | 3.07 ± 0.93 | 2.53 ± 0.71 | 3.84 ± 1.24 | 2.81 ± 0.59 | 3.62 ± 0.91 | 3.71 ± 0.86 | 50 |
| LFM-GNN (Lino et al. 2025) | 3.32 ± 1.06 | 2.87 ± 0.71 | 4.03 ± 1.34 | 2.89 ± 0.55 | 4.64 ± 1.14 | 4.07 ± 0.97 | 25 |
| FMT-8 | 1.67 ± 0.88 | **1.18 ± 0.40** | **2.71 ± 1.10** | **1.01 ± 0.27** | **2.29 ± 0.59** | 2.61 ± 0.83 | 20 |
| SAR (Ours) | **1.64 ± 0.70** | 1.51 ± 0.58 | 2.77 ± 1.11 | 1.37 ± 0.31 | 2.68 ± 0.81 | **2.23 ± 0.88** | 10+6+1 |

**Distributional Accuracy**  To evaluate how well the learned distributions replicate the ground-truth probability distribution derived from long simulations of fully developed flows, we measure their Wasserstein-2 ($W_2$) distance. The learned distributions are approximated by 200 samples in the ELLIPSE and ELLIPSEFLOW datasets and by 3,000 samples in the WING datasets. Table 1 reports $W_2$ distances for the ELLIPSEFLOW task under both in-distribution and out-of-distribution (OOD) variations in Reynolds number and geometric parameters. The first two subfigures in Figure 3 show the $W_2$ distances for the WING task, respectively, in a dataset built from training simulations

extended to the full distribution (WING-TRAINFULLDIST) and in a dataset containing unseen geometries (WING-INDIST). Across these two tasks, both the SAR and flow-matching Transolver models achieve substantially lower $W_2$ distances than the GNN baselines, despite the latter using fewer parameters (Table 3). We attribute this improvement primarily to the global receptive field of attention layers, which enables direct modeling of long-range spatial dependencies and full joint statistics of the flow field (Peebles & Xie, 2023). In contrast, multi-scale GNN architectures can only capture these dependencies after completing an entire sequence of message-passing operations across scales, making them less efficient at representing global interactions. For the smaller-scale ELLIPSE task, this advantage is less pronounced, as reflected in Table 4 for both in- and OOD settings.

Beyond using attention, SAR boosts distributional accuracy by decomposing generation into easier subproblems across resolution scales. Low-resolution predictions capture coarse global structures, while higher resolutions refine smaller-scale features. This hierarchy spares the model from capturing all variability at once, which likely explains why the largest SAR model (5.3M parameters) outperforms its Transolver counterparts on the WING task (Figure 3).

From a practical standpoint, improved distributional accuracy yields more reliable flow statistics. As shown in Figure 2b, SAR predicts turbulent kinetic energy (TKE)—involving the variance of velocity fluctuations—and Reynolds shear stress (RSS)—involving the covariance of these fluctuations—far better than the LDGN baseline on ELLIPSEFLOW-INDIST. While the flow-matching Transolver achieves slightly higher RSS accuracy, SAR is over six times faster (Figure 2 and 10), making it a compelling choice for fast, high-fidelity estimation of statistical flow quantities.

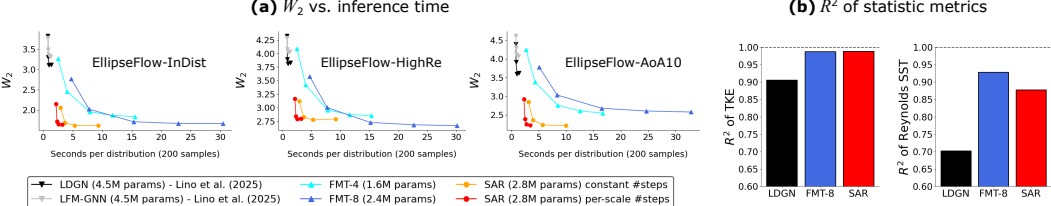

Figure 2: (a) Speed/distributional-accuracy trade-off on ELLIPSEFLOW-INDIST, ELLIPSEFLOW-HIGHRE, and ELLIPSEFLOW-AOA10. Curves for LDGN and LFM-GNN are obtained using 3, 5, 10, and 25 denoising steps. FMT curves use 3, 5, 10, 15, and 20 steps. The yellow SAR curve corresponds to using 2, 3, 5, and 10 denoising steps across all scales. The red SAR curve uses a different number of steps for each of the three scales: [2, 1, 1], [3, 2, 1], [5, 3, 1], and [10, 6, 1]. Inference times are measured on an NVIDIA RTX 3080. (b) Coefficient of determination ($R^2$) for Turbulent Kinetic Energy (TKE) and Reynolds Shear Stress (SST) on the ELLIPSEFLOW-INDIST dataset.

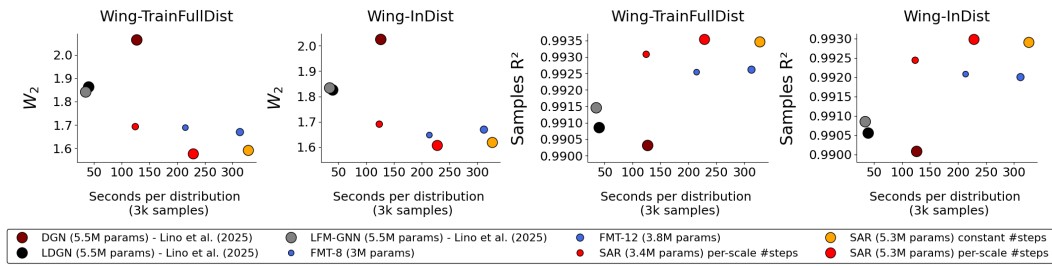

Figure 3: Speed and distributional/sample-accuracy trade-off on the WING-TRAINFULLDIST (training simulations extended to represent the full flow statistics) and WING-INDIST (design-space interpolation) datasets. Samples for DGN and LDGN are obtained using 5 denoising steps, and for LFM-GNN and FMT models using 3 steps. The yellow SAR values correspond to using 3 steps across all scales, while the red values correspond to 3, 2, and 1 steps in increasing resolution order. Inference times are measured on an NVIDIA RTX 3080. In this task, performance saturates quickly with the number of denoising steps. Because FMT models require very few steps, SAR's computational advantage is reduced; however, SAR still achieves superior accuracy.

**Sample Accuracy**   The quality of individual samples is also critical for obtaining physical insight or for use in downstream tasks (e.g., as initial conditions for numerical solvers). We approximate sample accuracy by comparing each generated state to all states from a simulated trajectory, selecting the one with the highest correlation, and reporting the corresponding coefficient of determination ($R^2$). For the ELLIPSE and ELLIPSEFLOW tasks, the trajectories are smooth and quasi-periodic, making $R^2$ a reliable indicator of sample accuracy. The WING dataset involves turbulent flows, which make difficult to align generated and ground-truth states. Nevertheless, we report the obtained $R^2$ values as a reference. Across all domains, SAR models consistently outperform GNN-based baselines, as shown by the $R^2$ values in Tables 5 (ELLIPSE datasets) and 6 (ELLIPSEFLOW datasets) and in the last to subfigures of Figure 3 (WING datasets). Visual comparisons are provided in Figure 8b for in-distribution samples in the ELLIPSE task and Figure 9 for OOD cases in the ELLIPSEFLOW task. We attribute the improved sample quality to the same factors underlying SAR's superior distributional accuracy: the global receptive field of attention layers and the scale decomposition.

**Computational Efficiency**   We evaluate the accuracy–runtime trade-off on the ELLIPSEFLOW and WING tasks (Figures 2, 3, and 11). While SAR can be run with a fixed number of denoising steps per scale, exploiting its scale-wise decomposition—allocating fewer steps to coarse scales and more to finer ones—proves significantly more efficient. In ELLIPSEFLOW (Figure 2), this adaptive strategy is over twice as fast as using a fixed step count. Under these optimized settings, SAR achieves $3$–$7\times$ faster inference than a flow-matching Transolver with 2.4M parameters and a number of denoising steps for which its accuracy is comparable or saturated. The challenging WING task, with its higher proportion of small-scale turbulent energy, causes the $W_2$ distance to saturate quickly with the number of denoising steps across all methods, reducing SAR's advantage. Even so, a 3.4M-parameter SAR is about $1.6\times$ faster than a 3M Transolver of similar accuracy, and SAR scales more favorably with size: the 5.3M SAR model improves over the 3.4M variant, whereas Transolvers show little benefit beyond 3M parameters.

Although on WING the $W_2$ distance saturates beyond three denoising steps per scale, the accuracy of the predicted standard deviation continues to improve up to 20 denoising steps, likely because it is a simpler metric reflecting only node-wise distributions. For this quantity, a SAR model using 20, 11, and 2 denoising steps (from coarser to finer scales) is $3\times$ faster than a flow-matching Transolver with 20 steps and similar accuracy (Figure 12b).

Finally, diffusion and flow-matching models based on multi-scale GNNs are $2.5\times$ faster on ELLIPSEFLOW and $5\times$ faster on WING due to their localized operations, but this speed comes at the cost of much poorer distributional and sample accuracy compared to SAR and Transolver models.

**Design Choices and Ablations**   SAR does not require a compressed latent space for efficiency. This is effectively equivalent to using a single denoising step at the highest-resolution scale, which we typically adopt. However, training in the latent space of a lightweight VAE proves beneficial: its decoder removes residual noise and corrects cross-scale misalignments, reducing the total denoising steps needed. As shown in Figure 4a for the ELLIPSE task, a non-latent SAR variant attains lower accuracy for the same step count and requires more steps to converge.

In image generation, some autoregressive models often predict multiple tokens per evaluation, but their probabilistic heads—e.g., linear layer followed by softmax (Tian et al., 2024) or diffusion-MLP (Li et al., 2024)—assign probabilities independently to each of them, which risks producing incompatible ouput features. We observed that replacing the Transolver-based sampler in SAR with a nodewise-MLP sampler causes severe degradation, as seen in the top-middle panel of Figure 4b for the ELLIPSE-INDIST dataset. By contrast, SAR's Transolver-based diffusion sampler explicitly models global spatial dependencies, yielding markedly higher sample quality.

The condition encoder plays a crucial role by providing global geometric and physical information to each node, regardless of scale. Without it, low-resolution predictions lack sufficient context, severely limiting generalization. For example, in the ELLIPSE-AOA10 dataset (ellipses at $10°$ angle of attack), omitting the encoder results in a clear drop in accuracy compared to the full SAR model (Figure 5a, right column), with visual differences in predicted fields and variances shown in Figure 5b. Although SAR uses Transolver-based components in our experiments, these can be replaced with alternative backbones, and, as more accurate or efficient architectures emerge, SAR can readily leverage them.

We also examine the effect of the number of scales. In ELLIPSEFLOW (Figure 5a, left), three scales give the best trade-off: multiple scales simplify the distribution, but more than three would demand

larger models due to parameter sharing across scales. A similar trend appears for ELLIPSE (Figure 8a). Finally, adding Gaussian noise with a standard deviation of $10^{-2}$ to lower-resolution inputs during training improves robustness to errors propagated from coarser scales (Figure 5a, middle).

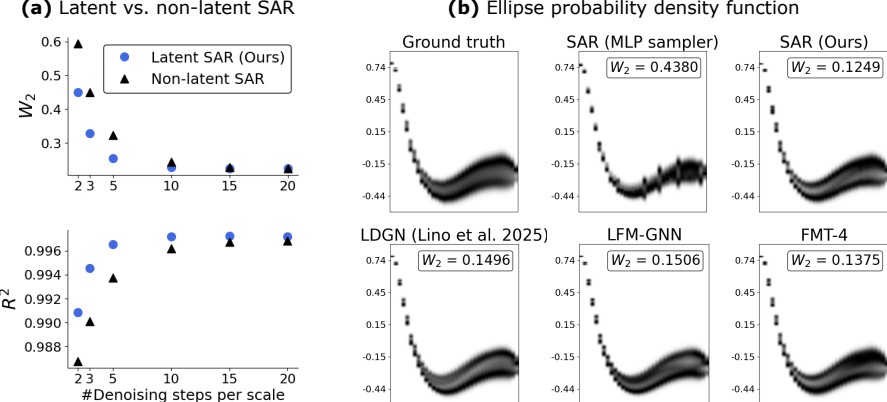

Figure 4: (a) Performance comparison between the latent SAR and non-latent SAR models across different numbers of denoising steps on the ELLIPSE-INDIST dataset. (b) Probability density function comparison for a sample from the ELLIPSE-INDIST dataset.

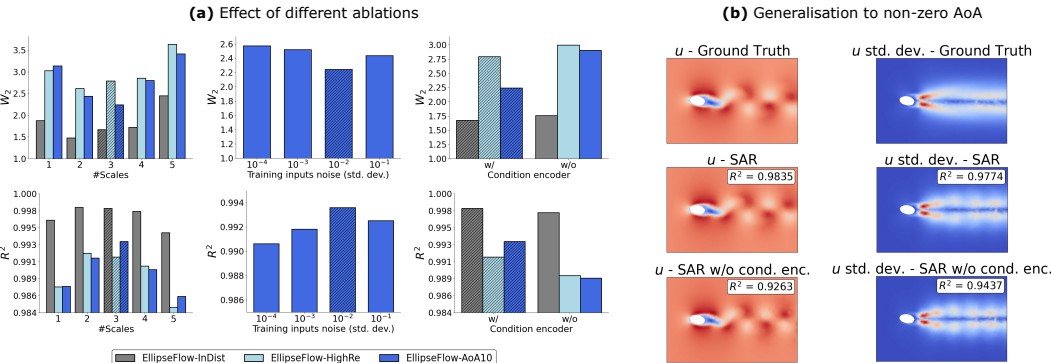

Figure 5: (a) Impact of ablation variants compared to our default configuration (striped bars), measured using Wasserstein-2 distance (top) and coefficient of determination (bottom). Bars report mean performance across the full test distributions from datasets ELLIPSEFLOW-INDIST, ELLIPSEFLOW-HIGHRE, and ELLIPSEFLOW-AOA10. (b) Visual comparison of SAR with (default) and without the condition encoder, shown on a representative sample from dataset ELLIPSEFLOW-AOA10.

## 5 CONCLUSIONS

We introduced *scale–autoregressive modeling* (SAR) for fluid flows on general geometries, factoring the joint distribution across resolutions and conditioning each finer scale on coarser predictions. This coarse-to-fine design concentrates denoising where uncertainty is highest and enables global-attention samplers with markedly fewer steps at high resolution. SAR achieves consistently lower distributional error and stronger sample accuracy than multi-scale GNN baselines, while matching or surpassing Transolver models with a more favorable accuracy–runtime trade-off. However, SAR has limitations that suggest promising directions for future work. In particular, it currently uses a fixed number of scales; making the hierarchy scale adaptive would better align accuracy and runtime with application preferences. In addition, exploring energy-based transformers (Gladstone et al., 2025) for the sampler could yield per-scale uncertainty estimates, enabling a more principled allocation of steps across scales. Despite these current limitations, we believe SAR is a compelling tool for fast, high-fidelity estimation of statistical flow quantities in real-world engineering applications.

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

# A  ADDITIONAL MODEL DETAILS

The implementation of our models and baselines, including their weights, and demonstration scripts are available at on-acceptance.

## A.1  SCALE ASSIGNMENT

SAR requires assigning a unique resolution scale $k$ (with $1 \leq k \leq K$) to each node $i \in \mathcal{V}$. To achieve this, we follow the procedure outlined in Algorithm 1, which ensures that nodes assigned to each scale are spatially distributed across the domain in a way that reflects the original mesh resolution—i.e., coarse regions remain coarse, and fine regions remain fine. This procedure iteratively applies Guillard coarsening (Guillard, 1993) to the original mesh-graph $\mathcal{G}$. At each coarsening step, a subset of nodes is retained to form a sparser graph $\mathcal{V}^{k+1}$, while the removed nodes are assigned the current finest available scale. To enable further coarsening, new edges must be defined for each $\mathcal{V}^{k+1}$. We reconstruct these edges by preserving the connectivity structure of the previous graph $\mathcal{G}^k$.

The resulting multi-scale node partition maintains the structural resolution hierarchy of the input mesh. Examples of the resulting node sets at different scales are shown in Figures 1a and 6.

---

**Algorithm 1** Guillard's coarsening algorithm (Guillard, 1993) and iterative scale assignment

```
 1: Γ ← ones(|V|)                                            ▷ Initialize scale assignment to 1
 2: for k ← 1 to K − 1 do
 3:
 4:     mask ← ones(|V^k|)                          ▷ Initialize the Guillard's coarsening mask
 5:     for node i ∈ V^k do                                         ▷ Iterate node-by-node
 6:         if mask[i] = 1 then            ▷ If first visit to node i then this node is not dropped
 7:             for node j ∈ N_i^- do
 8:                 mask[j] ← 0                             ▷ The incoming neighbours are dropped
 9:             end for
10:         end if
11:     end for
12:     V^{k+1} ← V^k[mask]            ▷ Drop the nodes based on the Guillard's coarsening mask
13:
14:     for node i ∈ V do                   ▷ Update the scale assigned to the non-dropped nodes
15:         if i ∈ V^{k+1} then
16:             γ_j ← k + 1
17:         end if
18:     end for
19:
20:     ... Create connectivity preserving edges edges (details omitted) ...
21:
22: end for
```

---

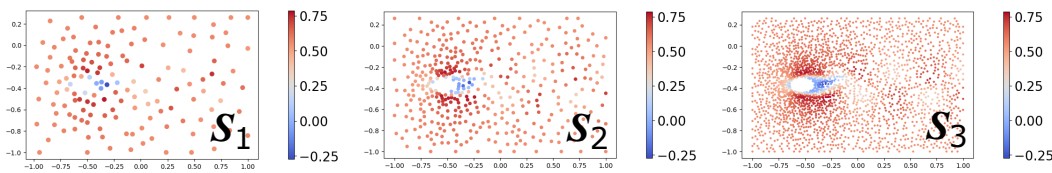

Figure 6: Horizontal component of the velocity field at each of the three resolution scales ($K = 3$) for a sample from the ELLIPSEFLOW-INDIST dataset.

## A.2  ARCHITECTURE DETAILS

The condition encoder, autoregressive module, and sampler components of our SAR model are all based on the Transolver architecture proposed in Wu et al. (2024), enhanced with adaptive temperature as introduced by Luo et al. (2025). This design provides a global receptive field efficiently by computing attention over a reduced set of *slice* tokens, each summarizing information from all

mesh nodes to some degree. We adopt Transolver due to its demonstrated effectiveness in modeling physical systems over general geometries (Wu et al., 2024; Luo et al., 2025).

A Transolver begins by projecting each input node feature vector $\boldsymbol{v}_i$ to the hidden dimensionality $F_{\text{model}}$ using a linear layer. Then, a series of Transolver blocks are applied, followed by a final projection to the desired output dimensionality. Each Transolver block contains a *physics-attention* layer and a node-wise MLP (Wu et al., 2024).

In the physics-attention layer, self-attention is applied independently across a set of learned latent representations, referred to as slices, for each attention head. This set of slices is denoted by $\mathcal{P}$. The number of slices $|\mathcal{P}|$ is relatively small compared to the total number of nodes $|\mathcal{V}|$ and remains fixed regardless of the graph size. As slice features are computed using a node-wise linear layer, the overall attention computation becomes effectively linear in the number of nodes.

In this layer, first, the node features are split into $H$ attention heads via a linear transformation:
$$[\boldsymbol{v}_i^1, \boldsymbol{v}_i^2, \dots, \boldsymbol{v}_i^h, \dots, \boldsymbol{v}_i^H] \leftarrow \text{LINEAR}(\boldsymbol{v}_i). \tag{5}$$
Then, for each head $h$ and node $i$, slice weights $\boldsymbol{w}_i^h \in \mathbb{R}^{|\mathcal{P}|}$ are computed as
$$\boldsymbol{w}_i^h \leftarrow \text{SOFTMAX}\left(\frac{\text{LINEAR}_h(\boldsymbol{v}_i^h)}{\tau_i^h}\right), \qquad \forall i \in \mathcal{V}. \tag{6}$$
Here, $\tau_i^h := \exp(\text{LINEAR}(\boldsymbol{v}_i^h)) \in \mathbb{R}^+$ is the adaptive temperature controlling the sharpness of the slice assignments (Luo et al., 2025). The weight $\boldsymbol{w}_i^h[j]$ denotes the degree to which node $i \in \mathcal{V}$ contributes to slice $j \in \mathcal{P}$ in head $h$.

Slice feature vectors are computed by weighted aggregation:
$$\boldsymbol{p}_j^h \leftarrow \frac{\sum_i^{|\mathcal{V}|} \boldsymbol{w}_i^h[j]\, \boldsymbol{v}_i^h}{\sum_i^{|V|} \boldsymbol{w}_i^h[j]}, \qquad \forall j \in \mathcal{P} \tag{7}$$

These slice features are then updated via standard unmasked self-attention (Vaswani et al., 2017). Afterward, the updated slice features are mapped back to the node space:
$$\boldsymbol{v}_i^h \leftarrow \sum_j^{|\mathcal{P}|} \boldsymbol{w}_i^h[j]\, \boldsymbol{p}_j^h, \qquad \forall i \in \mathcal{V}, \tag{8}$$

All heads are finally merged through a linear layer:
$$\boldsymbol{v}_i \leftarrow \text{LINEAR}\left([\boldsymbol{v}_i^1, \boldsymbol{v}_i^2, \dots, \boldsymbol{v}_i^h, \dots, \boldsymbol{v}_i^H]\right). \tag{9}$$

In the SAR condition encoder, we use Transolver blocks in their original form (Wu et al., 2024):
$$\boldsymbol{V} \leftarrow \boldsymbol{V} + \text{PHYSICS-ATTN}\left(\text{LAYERNORM}(\boldsymbol{V})\right), \tag{10}$$
$$\boldsymbol{V} \leftarrow \boldsymbol{V} + \text{MLP}\left(\text{LAYERNORM}(\boldsymbol{V})\right). \tag{11}$$

In the SAR autoregressive module and sampler, we further condition each Transolver block on the autoregressive step $k$ and the denoising step $r$, respectively. This is implemented using AdaLN-Zero—the adaptive layer normalization technique introduced by Peebles & Xie (2023) for diffusion transformer models—by modifying each block as follows:
$$[\boldsymbol{\alpha}, \boldsymbol{\beta}, \boldsymbol{\gamma}] \leftarrow [\boldsymbol{\alpha}_0^{\text{A}}, \boldsymbol{\beta}_0^{\text{A}}, \boldsymbol{\gamma}_0^{\text{A}}] + \text{MLP}(\text{EMB}) \tag{12}$$
$$\boldsymbol{V} \leftarrow \frac{\boldsymbol{V} - \text{LAYERMEAN}(\boldsymbol{V})}{\text{LAYERSTDDEV}(\boldsymbol{V})}\, \boldsymbol{\gamma} + \boldsymbol{\beta}, \tag{13}$$
$$\boldsymbol{V} \leftarrow \boldsymbol{V} + \boldsymbol{\alpha}\, \text{PHYSICS-ATTN}((\boldsymbol{V})), \tag{14}$$
$$[\boldsymbol{\alpha}, \boldsymbol{\beta}, \boldsymbol{\gamma}] \leftarrow [\boldsymbol{\alpha}_0^{\text{MLP}}, \boldsymbol{\beta}_0^{\text{MLP}}, \boldsymbol{\gamma}_0^{\text{MLP}}] + \text{MLP}(\text{EMB}) \tag{15}$$
$$\boldsymbol{V} \leftarrow \frac{\boldsymbol{V} - \text{LAYERMEAN}(\boldsymbol{V})}{\text{LAYERSTDDEV}(\boldsymbol{V})}\, \boldsymbol{\gamma} + \boldsymbol{\beta}, \tag{16}$$
$$\boldsymbol{V} \leftarrow \boldsymbol{V} + \boldsymbol{\alpha}\, \text{MLP}((\boldsymbol{V})), \tag{17}$$
$$\tag{18}$$

where EMB denotes the embedding of the scale or denoising time, and $\boldsymbol{\alpha}_0^{\square}, \boldsymbol{\beta}_0^{\square}, \boldsymbol{\gamma}_0^{\square}$ are learnable parameters—distinct for each block. The hidden size of each attention head in SAR is $F_{\text{model}}/H$, and every MLP has a single hidden layer with $F_{\text{model}}$ neurons.

**Condition Encoder**    The input feature vector for each node $i \in \mathcal{G}$ is constructed by concatenating its spatial coordinates $\boldsymbol{x}_i$, its conditioning features $\boldsymbol{v}_{c,i}$, and a one-hot vector $\boldsymbol{\gamma}_i$ encoding its assigned scale:

$$\boldsymbol{v}_i \leftarrow [\boldsymbol{x}_i, \boldsymbol{v}_{c,i}, \boldsymbol{\gamma}_i], \qquad \forall i \in \mathcal{V}.$$

These feature vectors are processed by the Transolver module (without AdaLN-Zero), producing latent representations $\boldsymbol{y}_i \in \mathbb{R}^{F_{\text{model}}}$ for each node.

**Autoregressive Module**    At autoregressive step $k$, we process all nodes belonging to the coarsest $k$ scales. For nodes in scales 1 through $k-1$, the input feature vector is constructed by concatenating a $F_{\text{model}}$-dimensional projection of the known (during training) or previously predicted (during inference) solution $\boldsymbol{s}_i \in \mathbb{R}^F$ and the latent vector $\boldsymbol{y}_i$:

$$\boldsymbol{v}_i \leftarrow [\text{LINEAR}(\boldsymbol{s}_i), \boldsymbol{y}_i], \qquad \forall i \in \mathcal{S}_{1:k-1}.$$

For nodes in the current scale $\mathcal{S}_k$, the input is a concatenation of a learnable mask embedding $\text{MASKEMB} \in \mathbb{R}^{F_{\text{model}}}$ (shared across iterations) and the latent vector $\boldsymbol{y}_i$:

$$\boldsymbol{v}_j \leftarrow [\text{MASKEMB}, \boldsymbol{y}_j], \qquad \forall j \in \mathcal{S}_k.$$

These inputs are processed by the Transolver variant with AdaLN-Zero. The AdaLN-Zero layers take as input a $F_{\text{model}}$-dimensional *iteration embedding*. Outputs corresponding to $\mathcal{S}_{1:k-1}$ nodes are ignored. The output features for nodes in $\mathcal{S}_k$ are denoted as $\boldsymbol{z}_j$.

**Sampler**    The sampler is a flow-matching model (Lipman et al., 2023) applied, at autoregressive step $k$, to nodes $j \in \mathcal{S}_k$. It is conditioned on each node's spatial location $\boldsymbol{x}_j$, scale one-hot vector $\boldsymbol{\gamma}_j$, and latent representations $\boldsymbol{y}_j$ and $\boldsymbol{z}_j$. The input feature vectors to the Transolver are defined as

$$\boldsymbol{v}_j \leftarrow [\text{MLP}([\boldsymbol{s}_{j,r}, \boldsymbol{z}_j, \text{MLP}([\boldsymbol{x}_j, \boldsymbol{\gamma}_j]) + \text{MLP}(\boldsymbol{y}_j)]), \boldsymbol{r}], \qquad \forall j \in \mathcal{S}_k, \tag{19}$$

where $\boldsymbol{s}_{j,r}$ is the intermediate solution at denoising time $r$, and $\boldsymbol{r} \in \mathbb{R}^{F_{\text{model}}}$ is the embedding of $r$. Given a scalar denoising time $r \in [0,1]$, its embedding vector is computed as

$$\boldsymbol{r} = \left[\sin(\omega_0\, r), \sin(\omega_1\, r), \ldots, \sin(\omega_{F_{\text{model}}/2-1}\, r), \cos(\omega_0\, r), \cos(\omega_1\, r), \ldots, \cos(\omega_{F_{\text{model}}/2-1}\, r)\right],$$

with

$$\omega_n = \exp\left(-\frac{\log(10000)}{F_{\text{model}}/2 - 1} \cdot n\right), \qquad n = 0, 1, \ldots, F_{\text{model}}/2 - 1.$$

The embedding vector $\boldsymbol{r}$ is also provided as input to the AdaLN-Zero layers.

**Variational Autoencoder (VAE)**    SAR operates in the latent space of a separately trained VAE, rather than directly in the physical space. This VAE is applied to the physical target fields defined on all nodes $\mathcal{V}$, and mesh-graph edges $\mathcal{E}$, where edge attributes $\boldsymbol{E}_c$ encode the relative positions between nodes. The architecture is compact, consisting of only two message-passing layers in both the encoder and the decoder.

These message-passing layers follow the framework described by Battaglia et al. (2016) and Battaglia et al. (2018). The edge- and node-update functions are modeled as single-hidden-layer MLPs with $F_{\text{model}}$ neurons and SELU activation functions using standard parameters (Klambauer et al., 2017). All MLPs are preceded by layer normalization (Ba et al., 2016). The steps are as follows:

$$\boldsymbol{e}_{ij} \leftarrow W_e \boldsymbol{e}_{ij} + \text{MLP}^e\left(\text{LN}\left([\boldsymbol{e}_{ij}|\boldsymbol{v}_i|\boldsymbol{v}_j]\right)\right), \qquad \forall (i,j) \in \mathcal{E}, \tag{20}$$

$$\bar{\boldsymbol{e}}_j \leftarrow \sum_{i \in \mathcal{N}_j^-} \boldsymbol{e}_{ij}, \qquad \forall j \in \mathcal{V}, \tag{21}$$

$$\boldsymbol{v}_j \leftarrow W_v \boldsymbol{v}_j + \text{MLP}^v\left(\text{LN}\left([\bar{\boldsymbol{e}}_j|\boldsymbol{v}_j]\right)\right), \qquad \forall j \in \mathcal{V}. \tag{22}$$

The VAE is trained to reconstruct the input node features with a low-weighted KL term between the latent distribution of each node, and a standard normal distribution:

$$\mathcal{L}_{\text{VAE}} = \frac{1}{|\mathcal{V}|} \sum_{i \in \mathcal{V}} ||\boldsymbol{z}_i - \boldsymbol{z}_i'||^2 + 10^{-6} \times \left(-\frac{1}{2|\mathcal{V}|} \sum_{i \in \mathcal{V}} \left(1 + \log\left(\sigma_i^2\right) - \mu_i^2 - \sigma_i^2\right)\right). \tag{23}$$

To enhance robustness against latent-space noise, Gaussian noise (with standard deviation 0.01) is introduced during VAE training. The initial learning rate is set to $10^{-4}$ and reduced by a factor of 10 when the training loss plateaus for a number of consecutive epochs: 50 for the ELLIPSE and ELLIPSEFLOW tasks, and 250 for the WING task (which uses shorter epochs). Training continues until the learning rate drops below $10^{-6}$.

The SAR backbone (condition encoder, autoregressive module, and sampler) is trained using the flow-matching loss described in Equation 4, and as detailed in Section 3.2.3. The initial learning rate is set to $10^{-3}$ and similarly reduced by a factor of 10 when the training loss plateaus for a number of consecutive epochs: 20 for the ELLIPSE and ELLIPSEFLOW tasks, and 100 for the WING task.

This training strategy is also applied to the flow-matching Transolver baselines, which can be considered equivalent to a single-scale, sampler-only SAR model.

### A.3 EXPERIMENTAL DETAILS

The diffusion graph network (DGN), latent DGN (LDGN), flow-matching GNN (FM-GNN), and latent FM-GNN (LFM-GNN) models used for each experimental domain are directly adopted from Lino et al. (2025). Our SAR and the flow-matching Transolver (FMT) baselines use only node-level conditioning features and do not incorporate edge conditioning, although this could be processed by the VAE as in Lino et al. (2025). We did not observe any reduction in accuracy due to this absence. The node conditioning features for each benchmark are summarized in Table 2.

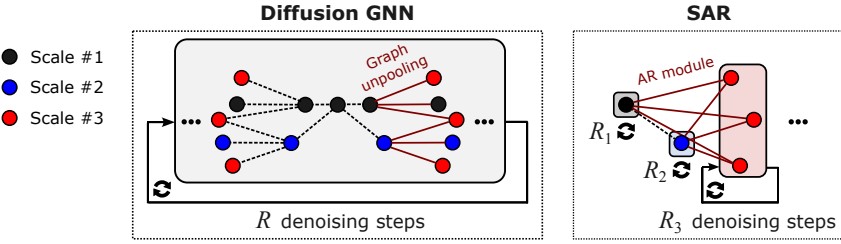

Figure 7: Diffusion GNN (Lino et al., 2025) and our SAR model both partition the node set $\mathcal{V}$ into resolution scales but differ in their processing approach. Diffusion GNNs apply the same number of denoising steps across all scales using local message passing and local unpooling. In contrast, SAR allows fewer denoising steps at finer scales, making it feasible to use otherwise expensive attention. Upsampling is performed once per scale via the transformer based *autoregressive module*.

Table 3 presents the total number of learnable parameters (combining VAE and backbone model) along with the hyperparameters for each model. Here, $F_{model}$ denotes the hidden size of node feature vectors (and edge feature vectors if available) in the backbone model, $F_{emb}$ is the size of the denoising-step or denoising-time embedding, $F_{VAE}$ represents the hidden size of the node and edge features in the VAE, $F_L$ is the dimensionality of the VAE latent space, $L_{cond}$ indicates the number of Transolver blocks in the condition encoder, $L_{AR}$ in the autoregressive module, and $L_{sampler}$ in the sampler.

Note that the VAE used in GNN-based models is a multi-scale GNN (Lino et al., 2022), while the VAE employed in the Transolver and SAR models is a flat GNN comprising two message-passing layers for the ELLIPSE and ELLIPSEFLOW tasks, and four layers for the WING task.

## B SUPPLEMENTARY RESULTS

Table 4 presents our measures of distributional accuracy, the graph-level Wasserstein-2 distance, on the ELLIPSE test datasets. Tables 5 and 6 report our measures of sample accuracy, the coefficient of determination, on the ELLIPSE and ELLIPSEFLOW test datasets, respectively. Figures 8b and 9 provide additional examples of results generated by SAR and baseline models for the ELLIPSE and ELLIPSEFLOW tasks.

Table 2: Conditioning node features and predicted outputs for each benchmark system.

| System | Node Condition Features ($v_{c,i}$) | Outputs ($s_i$) |
|---|---|---|
| ELLIPSE | Reynolds number ($Re$); distances to top and bottom walls | Surface pressure ($p_i$) |
| ELLIPSEFLOW | Reynolds number ($Re$); one-hot encoding of node type (inlet, ellipse boundary, interior) | Velocity components ($u_i$, $v_i$); pressure ($p_i$) |
| WING | Outward unit normal vector of the wing surface | Surface pressure ($p_i$) |

Table 3: Model size and hyperparameters.

| Task | Mode | #Params | $F_{model}$ | $F_{emb}$ | $F_{VAE}$ | $F_L$ | $L_{cond}$ | $L_{AR}$ | $L_{sampler}$ | #Scales |
|---|---|---|---|---|---|---|---|---|---|---|
| ELLIPSE | DGN | 3.51 M | 128 | 512 | – | – | – | – | – | 4 |
| | LDGN | 3.52 M | 128 | 512 | 126 | 1 | – | – | – | 2 + 2 |
| | FM-GNN | 3.51 M | 128 | 512 | – | – | – | – | – | 4 |
| | LFM-GNN | 3.52 M | 128 | 512 | 126 | 1 | – | – | – | 2 + 2 |
| | FMT-4 | 1.55 M | 128 | 128 | 128 | 1 | – | – | 4 | 1 |
| | SAR | 1.83 M | 128 | 128 | 128 | 1 | 2 | 2 | 2 | 3 |
| ELLIPSEFLOW | DGN | 4.50 M | 128 | 512 | – | – | – | – | – | 5 |
| | LDGN | 4.51 M | 128 | 512 | 126 | 1 | – | – | – | 2 + 3 |
| | LFM-GNN | 4.51 M | 128 | 512 | 126 | 1 | – | – | – | 2 + 3 |
| | FMT-4 | 1.55 M | 128 | 128 | 128 | 3 | – | – | 4 | 1 |
| | FMT-8 | 2.36 M | 128 | 128 | 128 | 3 | – | – | 8 | 1 |
| | SAR | 2.78 M | 128 | 128 | 128 | 3 | 4 | 4 | 4 | 3 |
| WING | DGN | 5.48 M | 128 | 512 | – | – | – | – | – | 6 |
| | LDGN | 5.49 M | 128 | 512 | 126 | 1 | – | – | – | 2 + 4 |
| | LFM-GNN | 5.49 M | 128 | 512 | 126 | 1 | – | – | – | 2 + 4 |
| | FMT-8 | 2.95 M | 128 | 128 | 128 | 1 | – | – | 8 | 1 |
| | FMT-12 | 3.75 M | 128 | 128 | 128 | 1 | – | – | 12 | 1 |
| | SAR-4x3 | 3.38 M | 128 | 128 | 128 | 1 | 4 | 4 | 4 | 3 |
| | SAR-4x8 | 5.25 M | 128 | 128 | 128 | 1 | 8 | 8 | 8 | 3 |

From a practical standpoint, improved distributional accuracy yields more reliable flow statistics. As shown in Figure 10, SAR predicts turbulent kinetic energy (TKE)—involving the variance of velocity fluctuations—and Reynolds shear stress (RSS)—involving the covariance of these fluctuations—far more accurately than the LDGN baseline on a simulation from ELLIPSEFLOW-INDIST. While the flow-matching Transolver achieves comparable accuracy, SAR is over six times faster.

Table 4: Wasserstein-2 distance ($W_2$) on the ELLIPSE datasets.

| ELLIPSE \ Model | -INDIST | -LOWRE | -HIGHRE | -THIN | -THICK | -AOA | #steps |
|---|---|---|---|---|---|---|---|
| DGN (Lino et al. 2025) | 0.29 ± 0.15 | 0.21 ± 0.09 | 0.42 ± 0.18 | 0.16 ± 0.02 | **0.56 ± 0.14** | 0.58 ± 0.10 | 50 |
| LDGN (Lino et al. 2025) | 0.23 ± 0.12 | 0.17 ± 0.08 | 0.42 ± 0.18 | **0.10 ± 0.02** | 0.57 ± 0.15 | 0.59 ± 0.11 | 50 |
| FM-GNN (Lino et al. 2025) | 0.31 ± 0.18 | 0.24 ± 0.18 | 0.46 ± 0.22 | 0.14 ± 0.03 | 0.68 ± 0.17 | 0.64 ± 0.09 | 10 |
| LFM-GNN (Lino et al. 2025) | 0.26 ± 0.14 | 0.19 ± 0.09 | 0.44 ± 0.18 | 0.12 ± 0.02 | 0.63 ± 0.17 | 0.61 ± 0.11 | 10 |
| FMT-4 | 0.29 ± 0.23 | 0.19 ± 0.15 | 0.44 ± 0.23 | 0.11 ± 0.03 | 0.67 ± 0.21 | 0.79 ± 0.14 | 20 |
| SAR (Ours) | **0.22 ± 0.12** | **0.15 ± 0.07** | **0.39 ± 0.17** | 0.11 ± 0.02 | 0.57 ± 0.18 | **0.57 ± 0.10** | 20 + 11 + 2 |

Table 5: Coefficient of determination ($R^2$) on the ELLIPSE datasets.

| ELLIPSE / Model | -INDIST | -LOWRE | -HIGHRE | -THIN | -THICK | -AOA | #steps |
|---|---|---|---|---|---|---|---|
| DGN (Lino et al. 2025) | $0.994 \pm 0.006$ | $0.997 \pm 0.001$ | $0.988 \pm 0.015$ | $0.994 \pm 0.002$ | $\underline{0.992 \pm 0.007}$ | $\mathbf{0.968 \pm 0.026}$ | 50 |
| LDGN (Lino et al. 2025) | $0.995 \pm 0.007$ | $0.998 \pm 0.002$ | $0.986 \pm 0.019$ | $0.997 \pm 0.001$ | $0.991 \pm 0.009$ | $\underline{0.966 \pm 0.028}$ | 50 |
| FM-GNN (Lino et al. 2025) | $0.995 \pm 0.007$ | $0.997 \pm 0.002$ | $0.987 \pm 0.015$ | $0.996 \pm 0.003$ | $0.991 \pm 0.009$ | $\underline{0.966 \pm 0.029}$ | 10 |
| LFM-GNN (Lino et al. 2025) | $0.995 \pm 0.008$ | $0.998 \pm 0.002$ | $0.985 \pm 0.020$ | $0.997 \pm 0.002$ | $0.990 \pm 0.011$ | $0.965 \pm 0.028$ | 10 |
| FMT-4 | $\mathbf{0.998 \pm 0.004}$ | $\mathbf{0.999 \pm 0.001}$ | $\mathbf{0.991 \pm 0.013}$ | $\mathbf{0.998 \pm 0.002}$ | $\mathbf{0.995 \pm 0.005}$ | $0.940 \pm 0.049$ | 20 |
| SAR (Ours) | $\underline{0.997 \pm 0.004}$ | $\mathbf{0.999 \pm 0.002}$ | $\mathbf{0.991 \pm 0.013}$ | $\mathbf{0.998 \pm 0.002}$ | $\underline{0.992 \pm 0.008}$ | $\underline{0.966 \pm 0.027}$ | 20+11+2 |

Table 6: Coefficient of determination ($R^2$) on the ELLIPSEFLOW datasets.

| ELLIPSE FLOW / Model | -INDIST | -LOWRE | -HIGHRE | -THIN | -THICK | -AOA | #steps |
|---|---|---|---|---|---|---|---|
| DGN (Lino et al. 2025) | $0.990 \pm 0.010$ | $0.993 \pm 0.007$ | $0.982 \pm 0.016$ | $0.989 \pm 0.009$ | $0.991 \pm 0.005$ | $0.987 \pm 0.014$ | 50 |
| LDGN (Lino et al. 2025) | $0.987 \pm 0.013$ | $0.992 \pm 0.009$ | $0.979 \pm 0.017$ | $0.986 \pm 0.011$ | $0.988 \pm 0.007$ | $0.981 \pm 0.016$ | 50 |
| LFM-GN (Lino et al. 2025) | $0.987 \pm 0.012$ | $0.992 \pm 0.008$ | $0.979 \pm 0.015$ | $0.985 \pm 0.010$ | $0.987 \pm 0.006$ | $0.983 \pm 0.014$ | 25 |
| FMT-8 (Lino et al. 2025) | $\mathbf{0.998 \pm 0.003}$ | $\mathbf{0.999 \pm 0.000}$ | $\underline{0.991 \pm 0.010}$ | $\mathbf{0.999 \pm 0.001}$ | $\mathbf{0.996 \pm 0.003}$ | $\underline{0.991 \pm 0.011}$ | 20 |
| SAR (Ours) | $\mathbf{0.998 \pm 0.003}$ | $\mathbf{0.999 \pm 0.001}$ | $\mathbf{0.992 \pm 0.008}$ | $\underline{0.998 \pm 0.002}$ | $\mathbf{0.996 \pm 0.003}$ | $\mathbf{0.994 \pm 0.009}$ | 10+6+1 |

We also evaluated the sample accuracy–runtime trade-off on the ELLIPSEFLOW task (Figure 11). As with distributional accuracy (Figure 2), SAR achieves 3–7× faster inference than a flow-matching Transolver with 2.4M parameters and a number of denoising steps for which its accuracy is comparable or already saturated.

Finally, although in the WING task the $W_2$ distance saturates beyond three denoising steps per scale (Figure 3), the accuracy of the predicted standard deviation continues to improve up to 20 denoising steps, likely because it is a simpler metric reflecting only node-wise distributions. For this quantity, a SAR model using 20, 11, and 2 denoising steps (from coarser to finer scales) is 3× faster than a flow-matching Transolver with 20 steps and similar accuracy, as illustrated in Figure 12b.

## C LLM USAGE

Parts of the final manuscript text were proofread and refined with assistance from OpenAI's ChatGPT. The model was used exclusively for language polishing at the paragraph level and was not employed for research ideation, experimental design, or retrieval and discovery tasks. The authors are solely responsible for all scientific content, claims, and conclusions presented in this paper.

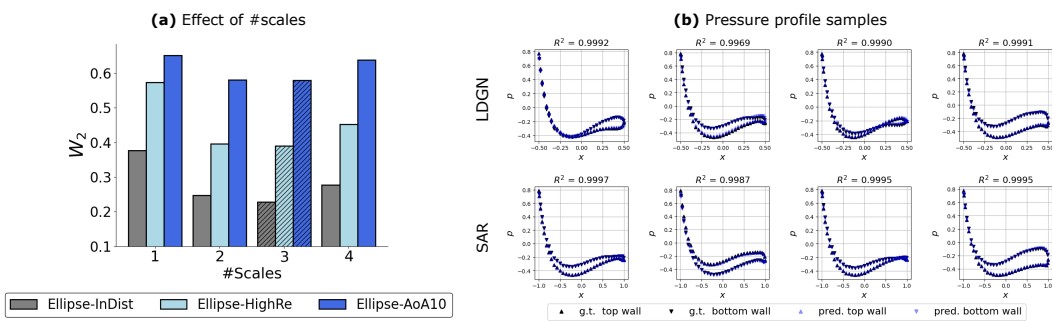

Figure 8: (a) Impact of the number of scales, measured using the Wasserstein-2 distance. Bars indicate mean performance across the full test distributions from the ELLIPSE-INDIST dataset. (b) Visual comparison of pressure profile samples predicted by SAR and LDGN (Lino et al., 2025) for an ellipse from ELLIPSE-INDIST with a relative thickness of 0.56 and $Re = 736$.

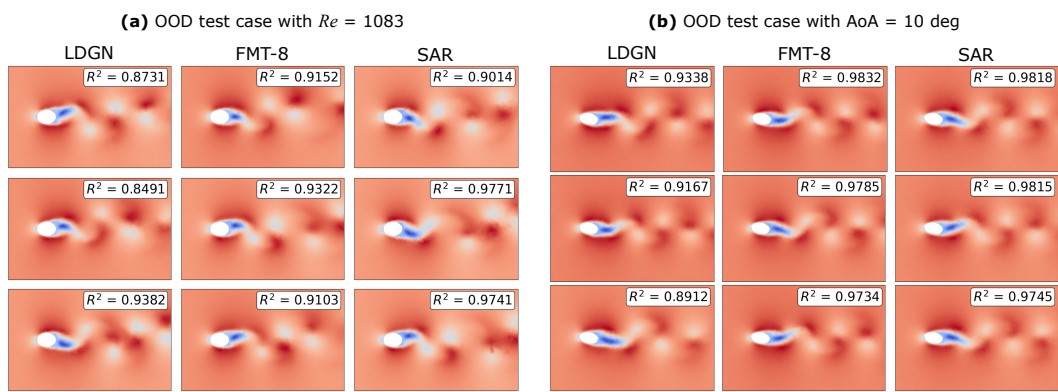

Figure 9: Samples from LDGN (Lino et al., 2025), FMT-8, and SAR for (a) a simulation from the ELLIPSEFLOW-HIGHRE dataset, and (b) a simulation from the ELLIPSEFLOW-AOA10 dataset. SAR produces the most accurate samples across both settings.

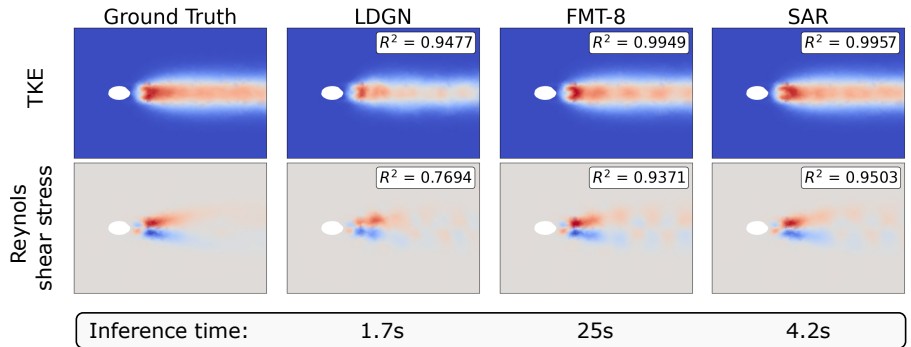

Figure 10: Turbulent kinetic energy (top row), Reynolds shear stress (middle row), and inference time (bottom row) for the distributions predicted by LDGN, FMT-8, and SAR for a test case from the ELLIPSEFLOW-INDIST dataset.

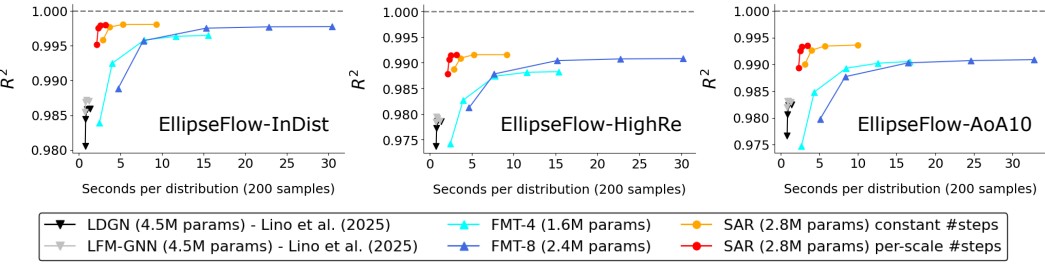

Figure 11: Speed/sample-accuracy trade-off on the ELLIPSEFLOW-INDIST, ELLIPSEFLOW-HIGHRE, and ELLIPSEFLOW-AOA10 datasets. Curves for LDGN and LFM-GNN are obtained using 3, 5, 10, and 25 denoising steps. FMT curves use 3, 5, 10, 15, and 20 steps. The yellow SAR curve corresponds to using 2, 3, 5, and 10 denoising steps across all scales. The red SAR curve uses a different number of steps for each of the three scales: [2, 1, 1], [3, 2, 1], [5, 3, 1], and [10, 6, 1]. Inference times are measured on an NVIDIA RTX 3080.

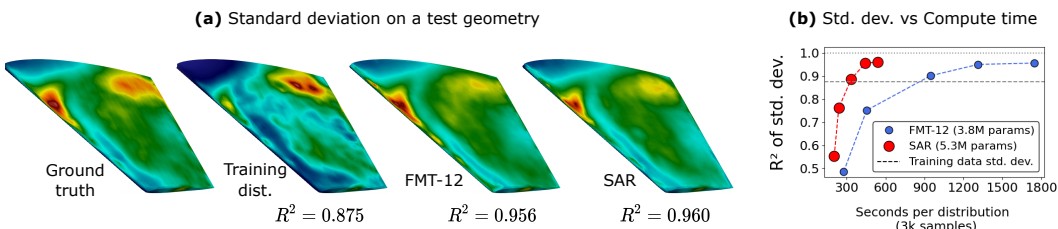

**(a)** Standard deviation on a test geometry

Ground truth

Training dist.    $R^2 = 0.875$

FMT-12    $R^2 = 0.956$

SAR    $R^2 = 0.960$

**(b)** Std. dev. vs Compute time

Figure 12: (a) Standard deviation of pressure on a wing geometry unseen during training (WING-INDIST dataset) from four sources: the ground-truth temporal distribution, the truncated training distribution, a flow-matching Transolver (FMT) model, and SAR. (b) For the same geometry, standard-deviation accuracy versus compute trade-off for the FMT and SAR models. FMT curves correspond to 3, 5, 10, 15, and 20 denoising steps. SAR curves use adaptive step configurations: [3, 2, 1], [5, 3, 1], [10, 3, 1], [15, 8, 2], and [20, 11, 2]. Inference times were measured on an NVIDIA RTX 3080.

