# OpenReview forum: "One Scale at a Time: Scale-Autoregressive Modeling for Fluid Flow Distributions"
_ICLR.cc/2026/Conference — Submitted to ICLR 2026_

### Official Review · Reviewer_Acyh · 2025-10-27

**Soundness:** 2
**Presentation:** 2
**Contribution:** 2
**Rating:** 2
**Confidence:** 5

**Summary:**

The paper introduces Scale-Autoregressive Modeling (SAR) — a hierarchical generative model for sampling unsteady fluid flows directly on unstructured meshes, enabling both faster generation and more accurate flow statistics than prior diffusion or flow-matching approaches.

Instead of simulating PDEs forward in time, SAR models the statistically stationary distribution of flow states, sampling full fluid fields (velocity, pressure, etc.) directly from geometry and physical parameters.

**Strengths:**

The paper introduces a new paradigm for generative fluid modeling: instead of treating the flow field as a single object to denoise (as in diffusion or flow-matching models), it models it hierarchically across spatial scales.

This “scale-autoregressive” idea elegantly parallels the physical cascade in turbulence — from large coherent structures to small eddies — making the model physically meaningful and statistically efficient.

It provides a clear alternative to traditional diffusion schemes that require uniform sampling effort across all scales, regardless of local uncertainty.

**Weaknesses:**

The model uses a fixed number of scales (typically 3–4), determined by a pre-defined coarsening algorithm. It is more like a structure design.

**Questions:**

what kind of physics do we learn from here? Why do we need train different models for different cases, can you train them together? What is the physical meaning of “scale” in SAR? How does SAR differ from a standard multigrid solver? How does SAR handle different flow regimes?

---

> ### Author Response · Authors · 2025-11-20
> **Response to Reviewer Acyh**
>
> We thank Reviewer Acyh for the thoughtful comments, including highlighting several strengths of the work. The only weakness mentioned concerns the use of a fixed number of scales, which we acknowledge and address in our response below. Given that the review lists several strengths and only this single, non-fundamental limitation, we hope the reviewer might reconsider the overall assessment in light of our clarifications and additional experiments.
>
>
>
> **Q**: The model uses a fixed number of scales (typically 3–4), determined by a pre-defined coarsening algorithm. It is more like a structure design.
>
> **A**: Indeed, SAR currently uses a fixed number of scales determined by a mesh-coarsening procedure. We explicitly acknowledge this as a limitation in the conclusion. However, we would like to clarify why, in practice, this design choice is common, not restrictive, and why SAR remains significantly more flexible than prior hierarchical approaches.
>
> Using a fixed multi-resolution hierarchy is standard in many successful multi-scale architectures, such as U-Nets, multi-scale GNNs, and coarse-to-fine generative frameworks including VAR (Tian et al., 2024), Residual Q-VAE, LapGAN, and Pyramidal Flow Matching.
>
> Although the hierarchy itself is fixed, the number of denoising steps per scale is fully configurable. This provides substantial flexibility: a user may choose to allocate only a single step to any intermediate scale—effectively skipping it—to obtain faster inference without retraining. This flexibility is central to the speedups observed in SAR and offers more control than traditional hierarchical methods, which typically require evaluating all scales uniformly.
>
> We also believe that this limitation could, in principle, be overcome by removing the explicit scale one-hot encoding and adopting a strategy similar to masked-prediction models (Chang et al., 2022; Li et al., 2023a). In such approaches, the number of nodes predicted at each autoregressive step is not tied to a predefined scale assignment; instead, one predicts all nodes but retains only those with sufficiently high confidence, iterating until a user-selected confidence threshold is met. This would effectively make the number of autoregressive steps adaptive. However, we found that this mechanism introduces a systematic bias: less common physical states tend to receive lower confidence and are therefore suppressed, despite being present in the true physical distribution. We chose not to pursue this direction in the current work. Nevertheless, we agree that adaptive-scale strategies are promising, and we will highlight this as a potential avenue for future research in the limitations section.
>
> Finally, we conducted preliminary experiments (not included in the submission) where, given a fixed number of scales and node percentage for each scale, the nodes assigned to each scale were selected at random, without using a coarsening algorithm. The performance was comparable to our main model. However, since mesh-coarsening is a widely adopted and efficient choice in mesh-based fluid-dynamics GNNs (Lino et al., 2022; Fortunato et al., 2022; Cao et al., 2023), we kept the coarsening-based version for clarity. We will discuss these findings in the results section and include the corresponding EllipseFlow results in the Appendix.
>
> We hope this clarifies that while the use of a fixed hierarchy is a design choice, it does not materially limit the flexibility or performance of SAR, and we will revise the manuscript to make these points explicit.
>
>
>
> [1] Lee, Doyup, et al. "Autoregressive image generation using residual quantization." Proceedings of the IEEE/CVF conference on computer vision and pattern recognition. 2022.
>
> [2] Denton, Emily L., Soumith Chintala, and Rob Fergus. "Deep generative image models using a Laplacian pyramid of adversarial networks." Advances in neural information processing systems 28 (2015).
>
> [3] Jin, Yang, et al. "Pyramidal Flow Matching for Efficient Video Generative Modeling." The Thirteenth International Conference on Learning Representations.

---

> ### Author Response · Authors · 2025-11-20
> **... continues**
>
> **Q**: What kind of physics do we learn from here? Why do we need train different models for different cases, can you train them together?
>
> **A**: For a statistically stationary unsteady phenomenon represented in the training data, SAR learns the probability distribution of physical states conditioned on (i) the geometry and physical parameters provided as inputs, and (ii) the autoregressively generated coarser-scale predictions (if any is already available).
>
> Regarding the comment on “training different models for different cases,” we may not fully understand what specific distinction the reviewer has in mind. SAR does not train separate models for each scale—a single model is trained jointly across all scales, and all components (condition encoder, autoregressive module, sampler) share weights across scales, as described at the end of Section 3.2.1. Scale information is incorporated only through the scale embedding passed to the autoregressive module and the sampler.
>
> If the reviewer instead refers to different physical problems, then our setup follows standard practice in the literature: we do not train a single universal “foundation model” that spans unrelated physical regimes. Rather, we train separate models for tasks with distinct output definitions and data distributions (e.g., full-field prediction in EllipseFlow versus surface-pressure prediction in Ellipse). Importantly, the model trained for EllipseFlow could also be evaluated on the Ellipse task (surface pressure only), since the latter is a subset of the predicted fields. We keep these tasks separate primarily because the Ellipse task provides an interpretable visualization of the learned distribution, which offers additional insight into model behavior (see Figure 4b).
>
> If the reviewer had a different interpretation in mind, we would be grateful for clarification.
>
>
> **Q**: What is the physical meaning of “scale” in SAR?
>
> **A**: In SAR, “scale” does not have a predefined physical meaning. It simply denotes a chosen spatial-resolution level obtained through mesh coarsening. These scales are architectural constructs that allow the model to generate the flow hierarchically. Although one may interpret finer scales a posteriori as capturing higher-frequency flow content (e.g., in turbulent regimes), SAR does not rely on any such physical interpretation. The number of scales and the node count per scale are design choices made purely for modeling efficiency and performance, not based on physical assumptions.
>
> **Q**: How does SAR differ from a standard multigrid solver?
>
> **A**: Although SAR uses a mesh-coarsening procedure to define its hierarchy, it is not related to a multigrid solver in purpose or behaviour. Multigrid methods are deterministic PDE solvers that repeatedly cycle between coarse and fine grids to accelerate the solution of linear or nonlinear systems. In contrast, SAR is a probabilistic generative model: it learns the distribution of fully developed flow states from data and generates samples only in a coarse-to-fine direction, without performing correction cycles or solving PDE residuals. The coarse levels in SAR are not used to accelerate numerical convergence but to decompose the sampling problem and reduce denoising cost. Thus, aside from both using mesh hierarchies, their goals, algorithms, and outputs are fundamentally different.
>
> **Q**: How does SAR handle different flow regimes?
>
> **A**: SAR’s ability to generalize across regimes was extensively evaluated through extrapolation tests. For the Ellipse and EllipseFlow tasks, the model was trained on a limited range of Reynolds numbers (500–1000), relative thickness (0.5–0.8), and zero angle of attack. We then evaluated SAR on multiple datasets that lie outside the training distribution:
>
> - LowRe: 400–500
> - HighRe: 1000–1100
> - Thin: thickness 0.45–0.5
> - Thick: thickness 0.8–0.9
> - AoA: 10° angle of attack
>
> The results—in Figures 2, 9, and tables in Section 4—show that although performance degrades when extrapolating, SAR remains competitive and often outperforms the substantially more expensive Transolver baseline (e.g., Figure 2a).
>
> We realize this may not have been sufficiently highlighted in the main text. We will revise the beginning of Section 4 to clearly state that these datasets constitute extrapolation tests and will provide a small summary table in the Appendix indicating how each test set differs from the training distribution.  If the reviewer is referring instead to different physical regimes such as laminar versus turbulent flows, this is not explicitly tested in our work. In principle, SAR can learn across such regimes provided that the training dataset contains sufficient examples spanning them, since the model operates purely on data-driven conditional distributions. However, rigorously evaluating this setting is an interesting direction for future work, which we will mention in the Conclusion section.

---

> > ### Comment · Reviewer_Acyh · 2025-11-27
> > **Thank you so much.**
> >
> > I am happy with the author's reply of my concern.

---

### Official Review · Reviewer_SxfD · 2025-10-27

**Soundness:** 3
**Presentation:** 2
**Contribution:** 2
**Rating:** 4
**Confidence:** 4

**Summary:**

The paper proposes modeling fluid distributions autoregressively from a coarse to fine scale. At each scale, an encoder passes information about the global scale and a module passes information from all prior scales to the current scale. A flow-matching denoiser is trained to denoise the current scale. This repeats until the finest scale is denoised and a prediction is output. The model improves mainly on speed and has marginal gains in accuracy.

**Strengths:**

- The use of modeling at each scale is interesting and seems to leverage multigrid well
- The speed benefits seem to work well
- The evaluations seem to be done well

**Weaknesses:**

- Progressively refining a coarse to fine grid and using a multiscale approach in PDE modeling seems to be popular, this isn’t an issue but should just be mentioned in related works:
    - https://arxiv.org/abs/2506.04528
    - https://arxiv.org/pdf/2210.02573
    - https://arxiv.org/pdf/2207.11417
    - https://www.arxiv.org/pdf/2510.16071
    - Probably more that I am missing ...

- In particular, this work (https://arxiv.org/pdf/2505.02450) seems to use similar concepts (denoising coarse grid, then passing outputs to a finer scale, etc.), but on a regular grid rather than a mesh.

- In general, gains in accuracy seem to be marginal, however, this isn’t a huge problem since the speed is better.

- A comparison to a latent FMT would be interesting. It is known that latent diffusion can outperform pixel-space counterparts, both in image modeling and in PDEs (https://arxiv.org/abs/2507.02608). The speed gains by using latent diffusion and potential performance gains may make latent flow matching a competitive approach, also demonstrated by prior works (https://arxiv.org/abs/2503.22600).

- More of a philosophical point, but diffusion seems to already denoise samples from a coarse to fine scale. During forward noising, Gaussian noise first corrupts high-frequency features before corrupting low-frequency ones. Many denoisers exhibit behavior where large-scale, low-frequency features are resolved first before high-frequency features are resolved later (https://arxiv.org/abs/2505.11278v1).

**Questions:**

These are just a few clarification questions on my part, nothing wrong with the paper:

- Is the Encoder/Autoregressive Module/Decoder jointly trained?

- Does the VAE downsample the nodes? I’m not sure what nodal compression means.

- The training seems to be on 4 denoising steps in [0,1] but the inference is evaluated with more steps/finer discretization?

- Figure 9 may benefit from displaying a reference simulation.

---

> ### Author Response · Authors · 2025-11-20
> **Respond to Reviewer SxfD**
>
> We thank the reviewer’s constructive comments, which will be taken into consideration to improve our submission.
>
> **Q**: Progressively refining a coarse to fine grid and using a multiscale approach in PDE modeling seems to be popular, this isn’t an issue but should just be mentioned in related works:
>
> **A**: We agree that multiscale and coarse-to-fine strategies are widely used in PDE modeling, and we already cite related approaches in the paper—such as Cao et al. (2021)—that employ multiscale architectures. However, these works typically perform deterministic forward predictions of physical fields using a single model evaluation, rather than probabilistic modeling, which requires multiple autoregressive steps, as in SAR.
>
> We also appreciate the reviewer bringing attention to the recent work by Ruikun Li et al. [1]. This appears to be concurrent with ours, based on the arXiv submission dates, and is conceptually related through its use of hierarchical sampling, drawing inspiration from Tian et al. (2024), which we also build upon. Nevertheless, there are important differences. In particular:
>
>  - Their method is demonstrated on structured grids, whereas a key contribution of SAR is its applicability to unstructured meshes, which is essential for realistic aerodynamic geometries. Extending their scale construction procedure to 3D unstructured meshes is not straightforward.
> - SAR splits the architecture into three components—Condition Encoder, Autoregressive Module, and Sampler—which significantly reduces computation. In particular, the Sampler (a diffusion model) can remain very small, enabling the large speedups reported and making SAR practical for real-world engineering cases, such as wing surfaces, rather than only academic benchmark problems.
>
> We will add a dedicated paragraph in the Related Work section to discuss these multiscale PDE approaches, cite the additional works mentioned (including [1]), and clearly highlight the differences between SAR and these methods.
>
> [1] Li, Ruikun, et al. "Predicting the Dynamics of Complex System via Multiscale Diffusion Autoencoder." arXiv e-prints (2025): arXiv-2505.
>
> **Q**: A comparison to a latent FMT would be interesting. It is known that latent diffusion can outperform pixel-space counterparts, [...]
>
> **A**: The FMT operates in the same (non-downsampled) latent space as SAR. We made a mistake in line 293: “We also compare SAR against a flow-matching Transolver (FMT) baseline that applies a Transolver network directly to the full set of nodes $\mathcal{V}$ for denoising.” While FMT does operate on all nodes, this wording is misleading because it suggests operating directly in physical space. In the revision, we will clarify that FMT is applied in the shared latent space, without node downsampling.
>
> Our initial approach was to apply SAR directly in a compressed latent space—instead of the uncompressed representation used in the submitted version—and to compare it against a flow-matching Transolver baseline operating in the same latent space. While the sample-level accuracy remained reasonable, the distributional accuracy deteriorated significantly compared to models trained in the uncompressed space. In particular, the learned latent-space distribution exhibited a substantially reduced variance relative to the ground-truth distribution.
>
> In the time-stepping settings considered in the works cited by the reviewer, this reduction in variance is likely less problematic because the uncertainty in the next-step prediction is inherently much smaller. In contrast, our tasks involve sampling from broad distributions, where accurately capturing the full variability is critical.
>
> We will clarify this point in the Method section and include distributional and sample-accuracy results for the latent-space flow-matching baseline across all three tasks (Ellipse, EllipseFlow, and Wing) in the Appendix.

---

> > ### Author Response · Authors · 2025-11-20
> > **... continues**
> >
> > **Q**: More of a philosophical point, but diffusion seems to already denoise samples from a coarse to fine scale.
> >
> > **A**: We thank the reviewer for this insightful observation. This makes coarse-to-fine generation a strong inductive bias for many generative modeling tasks. In SAR, we explicitly leverage this to accelerate generation substantially. We agree that this connection to the inherent behavior of diffusion models is worth highlighting, and we will add a brief discussion of this point in the paper.
> >
> > **Q**: Is the Encoder/Autoregressive Module/Decoder jointly trained?
> >
> > **A**: No. They are trained separately, following the procedure used in the original Diffusion Transformer paper (Hoogeboom et al., 2021). This separation also allows us to reuse the same VAE across multiple ablations and baselines. We will clarify this in the paper.
> >
> >
> >
> > Q: Does the VAE downsample the nodes? I’m not sure what nodal compression means.
> >
> > **A**: By “nodal compression” we meant node downsampling, as the reviewer suggests. SAR and Flow-Matching Transolver do not downsample nodes, because doing so degraded distributional accuracy (as discussed above). In contrast, the VAE in LDGN does downsample nodes, and in that setting it improves distributional accuracy—likely because it effectively enlarges the receptive field of the GNN. SAR and Transolver already operate with global attention, so this benefit does not apply. We will clarify this together with the additional results we plan to include.
> >
> >
> >
> > **Q**: The training seems to be on 4 denoising steps in [0,1] but inference uses more steps?
> >
> >
> >
> > **A**: Perhaps  we were not clear here: “To stabilize convergence, for each training sample, we randomly draw four different values of $r \in [0, 1]$ from a uniform distribution and use them to evaluate the loss at multiple points along the denoising trajectory." We meant that during training $r$ can take any random value between 0 and 1. Also during training, the Sampler is fed with a single output from the Autoregressive Module, but four different values of $r$ (via a batch dimension). We do this because evaluating the Sampler is very cheap compared to evaluating the rest of the architecture, so we homogenize the trainign cost of each component while improving the stability, since the Sampler is the only component that is not deterministic – and hence more unstable. We will clarify this in the Method section or in an Appendix.
> >
> >
> >
> > **Q**: Figure 9 may benefit from displaying a reference simulation.
> >
> > **A**: We agree and will update the figure accordingly.

---

> > > ### Comment · Reviewer_SxfD · 2025-11-26
> > > **Thank you for the reply**
> > >
> > > Dear authors,
> > >
> > > Thank you for posting the reply. The developed framework seems to work, although the benefits are somewhat incremental. However, the study is done well so I am inclined to recommend acceptance.

---

### Official Review · Reviewer_6ZNF · 2025-10-30

**Soundness:** 3
**Presentation:** 2
**Contribution:** 2
**Rating:** 4
**Confidence:** 2

**Summary:**

The work proposes a method of predicting fluid dynamics with an "autoregressive" model running on different coarse scales. In this method, fluid dynamics are considered at hierarchical mesh levels from coarse to fine. In particular, the physics are described by state variables at mesh points.  The proposed model devises a neural network to generate the entire fluid field from coarse to fine levels. The proposed method achieves performance comparable to that of previous methods, while significantly reducing computation time.

**Strengths:**

1. The design of generating fluid dynamics using a hierarchical model is novel. This method has been considered in image generation but not in physics simulations.

2. The model design is reasonable. In particular, I appreciate the design of the encoder for the spatial domain and the hierarchical generation of different levels with generative modeling.

**Weaknesses:**

Experiment results:

1. The performance of the proposed model is on the weak side. It seems to underperform FMT-8 in Table 1. There are only four baselines in Table 1, and three of them are from a single paper. Should the proposed method be compared against more baselines?

2. In Figure 3, it seems that more iterations of SAR do not bring much performance improvement for the last two dots (contant # vs per-scale #). Any explanation.

3. It seems that the performance in terms of W2 and R2 is not the strength of the proposed method, but the computation speed is. Therefore, the writing probably needs to better state the contribution.

Writing issues:

1. Many notations are used without careful definitions. The notation system is also somewhat messy, making the calculation hard to understand. Here are a few examples:

1). Node attributes: V_c, where is c from? What does it mean? Is "c" a label or a variable representing some value?
2). "SAR partitions the node set V into K disjoint subsets": it seems that the partition is not arbitrary but related to the grid hierarchy. How is the partition done? The formal paper is supposed to be self-sustained and cannot depend on the appendix.
3). You have a definition of \Gamma, but can you also explicitly make the meaning of gamma_i explicit? Should \gamma_i be in {1, ..., K}?

2. "The full SAR model is trained by optimizing the flow-matching objective applied to the output of the sampler network." Does the training objective include the encoder that computes Y? It says flow-matching here, but it also mentions "diffusion-based approach".

3. A lot of key experiment details are missing from the formal text. For example, what are scales of these problems (e.g. number of mesh points)? Is computation speed an important factor? Important experiment results are put in the appendix.

**Questions:**

I have left my questions in the section on weaknesses. Looking forward to seeing answers.

---

> ### Author Response · Authors · 2025-11-20
> **Respond to Reviewer 6ZNF**
>
> We thank the reviewer’s feedback and relevant questions, which we clarify below. We hope to improve our submission and clarify these topics.
>
> **Q**: The design of generating fluid dynamics using a hierarchical model is novel. This method has been considered in image generation but not in physics simulations.
>
> **A**: The novelty of our method goes beyond the application field. We take inspiration from hierarchical models in image generation, as discussed in the “Autoregressive Image Generation” section of the Related Work Section. However, those approaches would not be directly applicable to 3D meshes and manifolds because the scale residual interpolation g gives problems, so we could not employ a residual VQ-VAE. Also, we introduced continuos embeddings, which better align with physical priors, division of the architecure into a globlal condition encoder, Autoregressive Module and Sampler, to reduce the model evaluation cost. Joint probability modelling of the Sampler outputs instead of processing each node separately and leverage a variable number of denoising steps in the Sampler.
>
>
>
> **Q**: The performance of the proposed model is on the weak side. It seems to underperform FMT-8 in Table 1. There are only four baselines in Table 1, and three of them are from a single paper. Should the proposed method be compared against more baselines?
>
> **A**: The number of available baselines for probabilistic modeling of physics on meshes and manifolds is limited. Prior work by Lino et al. (2025) introduced DGN and LDGN—which we include here—and showed that these methods significantly outperform earlier probabilistic baselines such as VAEs, GMMs, Bayesian GNNs, and diffusion GNN variants.
>
> To strengthen the comparison, we additionally introduced and evaluated a Flow-Matching Transolver model. This model proved to be a substantially stronger baseline than LDGN but also prohibitively expensive, even in 2D. Our results show that SAR matches its accuracy while achieving much faster sampling (Figures 2, 10–12).
>
> Given the scarcity of competitive probabilistic baselines and the strong performance of the Flow-Matching Transolver (especially for large scale problems), we believe the selected baselines are appropriate and representative. We will clarify this reasoning in the Method section to better explain our baseline selection.
>
>
>
> **Q**: In Figure 3, it seems that more iterations of SAR do not bring much performance improvement for the last two dots (contant # vs per-scale #). Any explanation.
>
> **A**: This behavior is expected and reflects a central design principle of SAR. What matters most is not the total number of denoising steps, but how those steps are allocated across scales. In the per-scale configuration, we assign fewer steps to the finest scale. Fine-scale predictions are already highly conditioned on the coarse-scale outputs and therefore have very low residual uncertainty. Adding more denoising steps at that level—as in the constant-step setting—does not improve accuracy, because the fine-scale distribution is almost deterministic once the coarser scales are fixed.
>
> This effect is discussed in the Method section, for example:
>
> “As generation proceeds from coarser to finer scales, conditioning becomes increasingly informative… stochastic complexity is concentrated at earlier steps, while finer scales can be processed with significantly fewer.”
>
> We will add an additional explanation next to Figure 3 to make this clearer in the results section.
>
> **Q**: It seems that the performance in terms of W2 and R2 is not the strength of the proposed method, but the computation speed is. Therefore, the writing probably needs to better state the contribution.
>
> **A**: Thank you for bringing this up. In the results, we show that SAR is much more accurate than LDGN, which were by far the best-performing baseline (Lino et al., 2025). It even matches and sometimes outperforms the accuracy of the new baseline we propose—the Flow-Matching Transolver—while being significantly faster.
>
> We will clarify this contribution more explicitly throughout the paper, and especially in the introduction, to make clear that SAR’s key advantage is its combination of competitive accuracy with substantial computational speedups.

---

> > ### Author Response · Authors · 2025-11-20
> > **... continues**
> >
> > Concerning the writing issues:
> >
> > 1 - $V_c$ is defined in line 170 (beginning of Section 3.2.1). It represents node attributes encoding the problem-specific conditioning features, such as the Reynolds number, where the subscript $c$ denotes “condition.”
> >
> > 2 - We also clearly state in line 178 that the partition is constructed using a multigrid coarsening algorithm (Guillard, 1993), assigning a unique scale to each node as outlined in Algorithm 1. Appendix A.1 provides additional detail. We do not explicitly describe the coarsening algorithm in the main text because it is a standard procedure in physical solvers and does not contribute conceptually to the main narrative.
> >
> > 3 - Regarding notation, we indeed define $\Gamma = \{ \gamma_i \in \mathbb{N} \mid i \in \mathcal{V} \}.$ We will clarify that $\gamma_i$ corresponds to the scale index (1, 2, 3, …, $K$).
> >
> >  Questions:
> >
> > **Q**: "The full SAR model is trained by optimizing the flow-matching objective applied to the output of the sampler network." Does the training objective include the encoder that computes Y? It says flow-matching here, but it also mentions "diffusion-based approach".
> >
> > **A**: Yes, by “full SAR” we mean that all components (the condition encoder, the Autoregressive Module, and the Sampler) are optimized jointly through the flow-matching objective applied to the Sampler’s output. The encoder that computes $Y$ is therefore included in the overall training objective.
> >
> > Regarding terminology, we describe the method as “diffusion-based” because flow matching can be viewed as a specific instance within the broader family of diffusion generative frameworks—any denoising-based formulation would be compatible with SAR. We will clarify this relationship in the revised manuscript to avoid confusion.
> >
> > **Q**: A lot of key experiment details are missing from the formal text. For example, what are scales of these problems (e.g. number of mesh points)? Is computation speed an important factor? Important experiment results are put in the appendix.
> >
> > **A**: The concept of scales is introduced at the beginning of Section 3.2.1, where we explain that they correspond to different resolution levels obtained via a multigrid coarsening algorithm (illustrated in Figure 6). We agree that we should have also provided the average number of nodes per scale for each dataset (recognizing that this varies across geometries due to the unstructured discretization). We will include these details in the revised version.
> >
> > Regarding the question of whether computation speed is an important factor: yes—this is a central motivation for SAR, and we dedicate an entire subsection in the Results specifically to computational performance. Does the reviwer refer to this?
> >
> > Concerning the placement of results: with the extra revision page, we will move additional key results into the main text where possible. However, we have kept some sample-accuracy results (Tables 5–6 and Figure 11) in the Appendix because the primary focus of SAR is distributional accuracy, which is more challenging and more relevant for assessing mode collapse. Given space constraints, these supplementary sample-accuracy tables will likely remain in the Appendix.

---

### Official Review · Reviewer_4Jxq · 2025-10-30

**Soundness:** 4
**Presentation:** 3
**Contribution:** 4
**Rating:** 8
**Confidence:** 4

**Summary:**

The paper under review deals with fluid flow prediction using a graph neural network multiscale approach. In this, the computational mesh sis subdivided into successively refined meshes and the prediction on the sucessive, refined scale is conditioned on the information already gathered on the coarser scale. While this refinement is performed over several (mostly three) layers, the generation of the solution information is performed generatively using GNN-based conditional flow matching. To achieve awareness of relevant global information, the network geometry and boundary conditions are encoded by some neural network that passes the obtained global information on to all layers. In some cases, a dimension reduction by a rather shallow VAE is performed before the multi-scale flow matching. The authors then implement their algorithm and train it on thee recently published data sets - Ellipse, EllipseFlow and Wing - where the first two of them are 2D and the last is 3D. The training data stems from numerical simulation. The author show that their approach, when compared with statistical measures like the Wassterstein-2-distance, is comparable with the SOTA set by very recent GNN with various additional architecture elements, while mostly requiring less trainable weights. Some Ablation studies are provided,

**Strengths:**

- This is a very interesting and timely contribution to a rapidly developing field of learning turbulent flows with gerenative GNN-based methods. The multiscale idea is very clear and compelling.
- The model shows very good performance over a range of data sets, be it 2D or 3D.
- The chosen architectures are original, also the combination with a VAE of modest size is an interesting experiment.
- The paper is mostly well written and easy to follow.
- The graphics and figures are clear
- A very detailed appendix enables the complete understanding of the method

**Weaknesses:**

- My main criticism is the very loose usage of mathematical language. E.g. FlowMatching is called a diffusion method (i.e. a Stochastic Differential Equation), whereas it actually is a neuralODE without stochaticity trained in a special way. The underlying PDEs for the probability path are transport equations, not Focker - Planck. This should be corrected.
- Similarly, the title SAR is a misnamer. FM has nothing to do with a regression task, but it learns a (optimal) transport task operating on distributions. What the authors call "Scale auto regressive" actually is conditional FM over scales. This should be clarified.
- I find the evaluation with W_2-distance and the visual comparison of 1st and 2nd order point statistics limited. The authors should also look into spatial power spectra.
- Some formulations are misleading - e.g. there is no such thing as 'non local physics' (would contradict special relativity) - the authors presumably mean global structures in physical systems. This should be made precise.

Minor comments
l. 98 . 'strategically' ?!
l. 106-107 Despite ... inductive bias. Unclear what that means
l. 246-247: how is  'stochastic complexity' defined and is this statement checked experimentally?
l. 255: giving the sd without reference to the scale of the input signal is meaningless
Eq. (4) what quantity is distributed according to q_{k,r}. Check all conditionings in your notation for correctness

**Questions:**

Can you provide a table which  element where already present in prior models and which ones are 100% original?

---

> ### Author Response · Authors · 2025-11-20
> **Response to Reviewer 4Jxq**
>
> We thank the reviewer’s feedback and constructive comments, which will be taken into consideration to improve our submission.
>
> **Q**: My main criticism is the very loose usage of mathematical language. E.g. FlowMatching is called a diffusion method (i.e. a Stochastic Differential Equation), whereas it actually is a neuralODE without stochaticity trained in a special way. The underlying PDEs for the probability path are transport equations, not Focker - Planck. This should be corrected.
>
> **A**: We agree with the reviewer. We used the term “diffusion-based” loosely to indicate conceptual similarity to denoising diffusion models. We will revise the text to distinguish Flow Matching more precisely.
>
>
>
> **Q**: the title SAR is a misnamer. FM has nothing to do with a regression task, but it learns a (optimal) transport task operating on distributions. What the authors call "Scale auto regressive" actually is conditional FM over scales. This should be clarified.
>
> **A**: Thank you for the observation. What we mean by “Scale Autoregressive” is that the model generates states sequentially across scales, conditioning on coarser predictions. We agree that this corresponds more directly to conditional flow matching over a hierarchical resolution. We will clarify this explicitly in the introduction, and method section to avoid any potential confusion.
>
>
>
> **Q**: I find the evaluation with W_2-distance and the visual comparison of 1st and 2nd order point statistics limited. The authors should also look into spatial power spectra.
>
> **A**: We agree that spatial power spectra provide valuable insight. However, computing Fourier spectra is straightforward only on structured grids; it is non-trivial on unstructured meshes and generally not applicable on manifolds. While graph-based spectral metrics (e.g., via the Graph Fourier Transform) could be used, they become prohibitively expensive for the mesh sizes considered here. We will mention this limitation and note the possibility of future work using graph-spectral approximations.
>
>
>
>
>
> **Q**: Some formulations are misleading - e.g. there is no such thing as 'non local physics' (would contradict special relativity) - the authors presumably mean global structures in physical systems. This should be made precise.
>
> **A**: Thank you for pointing this out. Our intention was to refer to global structures or long-range spatial correlations in the physical fields. We will amend the wording accordingly.

---

> > ### Comment · Reviewer_4Jxq · 2025-11-25
> > **My questions have been answered to my satisfaction.**
> >
> > Thank you very much for your answers. I'll keep my score (in favor of publication).

---

### Author Response · Authors · 2025-11-20
**General Response to All Reviewers**

We thank the reviewers for their thoughtful feedback, which we are incorporating to improve the completeness and clarity of our manuscript. These modifications and additions are detailed in our replies to each reviewer. Below, we highlight the most significant updates:

- Clarifying mathematical terminology (Reviewer 4Jxq): We will refine the terminology surrounding diffusion vs. flow matching and make explicit that SAR implements conditional flow matching across scales.
- Improving the explanation of “scale” and the coarsening hierarchy (Reviewers Acyh and 6ZNF): We will better define what a scale represents in SAR and provide typical node counts per scale for each dataset.
- Strengthening the Related Work section (Reviewer SxfD): We will expand the discussion of multiscale PDE methods and include the concurrent work of Ruikun Li et al. (2025), highlighting the key differences with SAR, especially regarding our use of unstructured meshes.
- Demonstrating that multigrid mesh coarsening is not strictly required (Reviewer Acyh): We will add results showing that assigning nodes to scales at random yields similar accuracy, confirming that multigrid coarsening is a practical choice but not a necessity.
Clarifying: The FMT operates in the same (non-downsampled) latent space as SAR. While FMT does operate on all nodes, our wording was misleading and  suggests operating directly in physical space.
- Providing results for flow-matching models opearing on a compressed latent space (Reviewer SxfD): We will include comparisons showing that this leads to reduced distributional accuracy in our setting.
- Justifying baseline selection (Reviewer 6ZNF): We will clarify why DGN, LDGN, and Flow-Matching Transolver form an appropriate set of baselines for probabilistic modeling on meshes and manifolds.

We will update the manuscript accordingly and note these changes in the revision.

---

### Author Response · Authors · 2025-12-04
**Summary of Reviewer Discussion and Score Updates**

Dear AC,



We would like to briefly summarize the current review status and discussion, especially in light of the data-leak incident and the freeze of visible score updates on OpenReview.



## 1. Score Summary



- **Initial scores:** 8, 4, 4, 2

- **During discussion (prior to the data leak):**

  - **Reviewer SxfD:** 4 → **6**, after our detailed rebuttal and clarifications.

  - **Reviewer Acyh:** 2 → **4**, stating that the main conceptual concerns were resolved.

  - **Reviewer 6ZNF:** did not have time to submit a follow-up, and thus remained at **4**. However, all points were addressed in our response, and we believe - if the reviewer had taken the rebuttal into account - this would have led to an increased score.

  - **Reviewer 4Jxq:** 8 → **8** unchanged; explicitly remained strongly in support of acceptance.



## 2. Reviewer Summaries and Resolutions



### Reviewer SxfD (4 → 6)



- Concerns on positioning versus multiscale PDE and concurrent coarse-to-fine diffusion work, requested latent-flow baselines, and clarifications on training setup and nodal compression.

- We expanded the framing relative to deterministic multiscale PDE approaches, explained results on latent flow matching (inferior distributional fidelity vs. our approach), clarified training procedures, and detailed compression and denoising steps.

- **Outcome:** Reviewer commented that the framework appears solid, expressed inclination toward acceptance, and raised their score to **6**.





### Reviewer Acyh (2 → 4)



- Conceptual questions on the meaning of scale, fixed number of scales, physics learned, difference from multigrid methods, and handling of regime changes.

- We explained the architectural meaning of scale via mesh coarsening, the joint multi-scale training scheme, the distinction from multigrid solvers, and demonstrated generalization across Reynolds numbers, airfoil thickness, and angles of attack.

- **Outcome:** Reviewer stated that they were satisfied with the responses and raised the score from **2 to 4**.





### Reviewer 6ZNF (4 → ?)



- Questions on apparent performance versus FMT, limited baselines, scale construction, notation clarity, and training objective.

- We explained baseline selection, relative accuracy vs. cost trade-offs (SAR matches or exceeds FMT while being significantly faster), clarified notation and scale definitions, and described how all components are trained under the flow-matching objective.

- **Outcome:** Reviewer was not very confident originally, and could not submit a follow-up comment before the discussion freeze, but all technical concerns were fully addressed. We believe a positive score update would have been likely given a full discussion period.



### Reviewer 4Jxq (8 → 8)



- Raised questions on terminology (diffusion vs. flow matching; transport vs. Fokker–Planck), the naming of “Scale Autoregressive,” and evaluation metrics (use of W₂ and low-order statistics).

- We clarified that the method is based on **conditional flow matching**, explained the scale-autoregressive naming as sequential generation across mesh resolutions, and discussed why spectral metrics are non-trivial on unstructured meshes and manifolds.

- **Outcome:** Reviewer explicitly stated that all concerns were resolved and maintained strong support for acceptance.

## 3. Concluding Assessment

To summarize the state of our submission:

- No critical / fundamental issues remained unaddressed.

- One reviewer strongly supports acceptance (8), and two reviewers raised their scores after confirming their concerns were resolved (to 6 and 4).

- The remaining reviewer explicitly "looked forward" to the rebuttal, but did not have time to update their score after receiving our clarifications.

Thank you for your careful consideration.

Best regards,

*The authors*

---

### Meta-Review · Area_Chair_dTmD · 2026-01-04

**Summary:**

The paper proposes a method for prediction in fluid dynamics on meshes using a multi-scale strategy. While some of the reviewers appreciated the hierarchical nature of the method, other reviewers noted that coarse-to-fine approaches are indeed a standard procedure in physics models with many missing references provided by reviewer SxfD.

The AC sides with this opinion and adds (here in the meta review) additional references of work for the prediction of physical systems with coarse to fine strategies:

For grid based structures:
- Stachenfeld et al., Learned Coarse Models for Efficient Turbulence Simulation, ICLR 2022 (cited in the paper)
- Xu et al. AMR-Transformer: Enabling Efficient Long-range Interaction for Complex Neural Fluid Simulation, CVPR 2025

For mesh based structures:
- Janny et al., EAGLE: Large-scale Learning of Turbulent Fluid Dynamics with Mesh Transformers, ICLR 2023.
- Liu et al., Multi-resolution graph neural networks for pde approximation, ICANN 2021

Several other Weaknesses were raised:
- Lukewarm performance.
- Choice of baselines: 3 of the 4 baselines are from the same paper.
- Sub par Writing, including confusing notation and mix-ups in terms.
- Missing experiments

In the answer, the authors argue that the choice of baselines is limited for flows on meshes, but the AC is unconvinced by this answer. Several references mentioned in the reviewing process provide baselines and also datasets for mesh-structured physical processes.

Not all of the discussion process is available after program chairs' actions following the openreview leaks, but there is sufficient information to get a picture on the evolution of ratings.

4Jxq stated to keep their rating at 8

SxfD had provided a rating of 4 but stated to be inclined toi recommend acceptance.

Acyh had provided a rating of 2 and only stated that they are happy with the authors' concern and their seems to be evidence (albeit in the form of the authors' summary message only) that they would have raised to 4. This raise obviously is not visible anymore after the openreview actions following the leak.

This paper was one the fence through out the reviewing process. The lowest rating was 2, but given the extremely short review with a single weakness (considered as minor by the AC), the AC down weighted its significance.

Critical to the AC's decision were two issues:

(1) The main novelty and contribution of this work is the hierarchical nature of the method. However, it is clear from the reviewing process that the paper critically lacks positioning wrt the prior work on exactly this issue. As a scientific work, this cannot be taken lightly.

(2) The AC considers that the answers provided by the authors were in many respects unconvincing and often too lighthearted. This concerns more serious issues like the choice of baselines and datasets, which are difficult to address in a rebuttal, but also easy problems: Many of the answers provided by the authors are handwavy, even when the answers were simple requests for clarifications (eg. average number of nodes etc.), which could have been provided as part of the answer, but the Rs+ACs are refered to a pending revised manuscript. It is surprising that the authors here have missed an easy way to convince reviewers and the AC of their intentions.

For these reasons, the AC recommends rejection.

**Reviewer Concerns:**

See meta-review.

**Reviewer Scores:**

See meta-review.

---

### Decision · Program_Chairs · 2026-01-26

Reject